# Scaling Quantum Machine Learning without Tricks: High-Resolution and Diverse Image Generation

## Abstract

Quantum generative modeling is a rapidly evolving discipline at the intersection of quantum computing and machine learning. Contemporary quantum machine learning is generally limited to toy examples or heavily restricted datasets with few elements. This is not only due to the current limitations of available quantum hardware but also due to the absence of inductive biases arising from application-agnostic designs. Current quantum solutions must resort to *tricks* to scale down high-resolution images, such as relying heavily on dimensionality reduction or utilizing multiple quantum models for low-resolution image patches. Building on recent developments in classical image loading to quantum computers, we circumvent these limitations and train quantum Wasserstein GANs on the established classical MNIST and Fashion-MNIST datasets. Using the complete datasets, our system generates full-resolution images across all ten classes and establishes a new state-of-the-art performance with a single end-to-end quantum generator without tricks. As a proof-of-principle, we also demonstrate that our approach can be extended to color images, exemplified on the Street View House Numbers dataset. We analyze how the choice of variational circuit architecture introduces inductive biases, which crucially unlock this performance. Furthermore, enhanced noise input techniques enable highly diverse image generation while maintaining quality. Finally, we show promising results even under quantum shot noise conditions.

## 1 Introduction

Since the advent of ChatGPT (OpenAI, 2025), generative modeling has become one of the most used technologies in the world (Paris, 2023). From coding "copilots" (Yao, 2023) to the generation of realistic-looking images (OpenAI, 2022), or musical compositions (OpenAI, 2019), generative AI is continuously gaining fields of applications, with increasing computation and energy demands (Jegham et al., 2025). Quantum generative modeling (Schuld & Petruccione, 2021) is an emerging field at the intersection of quantum computing and machine learning, focused on using quantum systems to learn, model, and sample from complex data distributions. Just as classical generative models, e.g. Variational Autoencoders (VAEs) (Kingma & Welling, 2014), Generative Adversarial Networks (GANs) (Goodfellow et al., 2014), or Transformers (Vaswani et al., 2017), learn to mimic data distributions, quantum generative models aim to leverage the probabilistic and high-dimensional nature of quantum mechanics to achieve outcomes, potentially superior and intractable for classical computers (Huang et al., 2025). Although the potential advantages of applying quantum generative models to practical problems remain uncertain in terms of performance, there are indications that such systems can be energetically more efficient (Villalonga et al., 2020). Thus, it is crucial to investigate their capabilities on relevant machine learning benchmark tasks empirically.

Image generation is a particularly interesting use case of generative modeling. For example, data augmentation (Islam et al., 2024) for artificial vision systems is used in diverse fields ranging from medical diagnose systems (Motamed et al., 2021) to quality assurance (Wang et al., 2023), in which neural networks are trained to recognize illness or defective parts or products. In both cases, such anomalous images are usually difficult to obtain naturally and synthetic examples need to be created.

State-of-the-art methods for quantum image generation rely on *tricks* to circumvent scaling issues related to high-dimensional (high-resolution) images. We recognize two widely used techniques:

1. *Dimensionality reduction:* This method uses principal component analysis (PCA) (Stein et al., 2021; Silver et al., 2023; Chu et al., 2023; Solanki et al., 2024; Khatun et al., 2024) or neural networks, including autoencoders, (Rudolph et al., 2022; J et al., 2022; Shu et al., 2024; Ma et al., 2025) to generate images in a lower-dimensional *latent* space. The output of the small quantum model is then classically post-processed to recover the original image dimensions.

2. *Patch generation:* This method circumvents high dimensionality by generating smaller patches of the images, where each patch uses a separate quantum generator, usually trained simultaneously (Huang et al., 2021; Tsang et al., 2023; Thomas & Jose, 2024).

Importantly, both methods circumvent high-dimensional data by generating low-dimensional quantum model outputs and may supplement them with classical computation to recover the original image dimensions. As a result, it becomes unclear whether the quantum model plays a non-trivial role in the generation. This is particularly true for the first method type, where a neural network may cover most of the generation. Thus, we consider Tsang et al. (2023), a patch-generation QGAN with one quantum generator per image row, as the previous state-of-the-art and baseline for comparison. Notably, despite these tricks, prior QGANs suffered from limited visual quality and diversity, producing scattered pixels and unrealistic class mixing even on three-class datasets. By presenting a single end-to-end quantum generator for diverse images at full resolution, we provide evidence for the capability and scalability of quantum generative modeling when appropriately designed.

Data of interest are often not arbitrary and have some internal structure, e.g., natural occurring images differ from random pixels. In fact, real images are known to have low-rank structure, evidenced in their fast decreasing power spectrum (van der Schaaf & van Hateren, 1996). This allows for compression algorithms such as JPEG (Wallace, 1992), which is a popular format in classical computing. This structure carries out to the quantum realm, as illustrated in several recent results (both numerical and theoretical) showing that their underlying structure leads to encoding quantum states that are well-captured by tensor-network states and by tensor-network-inspired quantum circuits (Dilip et al., 2022; Iaconis & Johri, 2023; Jobst et al., 2024; Shen et al., 2024). These states can thus be prepared with quantum circuits of depth linear in the number of qubits required for the encoding.

Prior research has explored various aspects of quantum image processing, including the identification of effective quantum encodings (Jobst et al., 2024), the generation of large-scale datasets through quantum circuit-based image encoding (Kiwit et al., 2025), and the application of quantum models to classification tasks (Shen et al., 2024; Kiwit et al., 2025). Here, we present a single end-to-end image quantum generator based on a quantum GAN (QGAN) training with a classical discriminator. In our approach, we use no dimensionality reduction methods and no multiple generators for image patches, and tackle large datasets commonly used in the machine learning field for benchmarking: MNIST (Lecun et al., 1998), Fashion-MNIST (Xiao et al., 2017), and Street View House Numbers (SVHN) (Netzer et al., 2011), for color images. This is possible due to the inductive bias created by an application-specific quantum circuit design inspired by the exponentially compressed encoding scheme. Moreover, we show that multimodal noise input increases the diversity of the generated images. We further explore the performance of training in the presence of shot noise.

## 2 BACKGROUND: QUANTUM IMAGE REPRESENTATIONS

The simplest way to encode classical data into the amplitudes of a quantum state is referred to as *amplitude encoding* that is given by $|\psi(\boldsymbol{x})\rangle = \frac{1}{\|\boldsymbol{x}\|} \sum_{j=0}^{2^A-1} x_j |j\rangle$, where $\boldsymbol{x}$ represents some classical data vector (Schuld & Petruccione, 2021; Latorre, 2005). (For notation conventions, see App. A). This encoding is attractive because it allows for representing an image with $2^A$ pixels using only $A$ qubits, leading to an exponential reduction in storage requirements compared to a classical representation. Since the state must be normalized, the global scaling information is lost in the encoding. To address this limitation, encodings of the following form have been proposed (Le et al., 2011a;b) for images with $2^A$ pixels:

$$|\psi(\boldsymbol{x})\rangle = \frac{1}{\sqrt{2^A}} \sum_{j=0}^{2^A-1} |c(\boldsymbol{x}_j)\rangle \otimes |j\rangle .\tag{1}$$

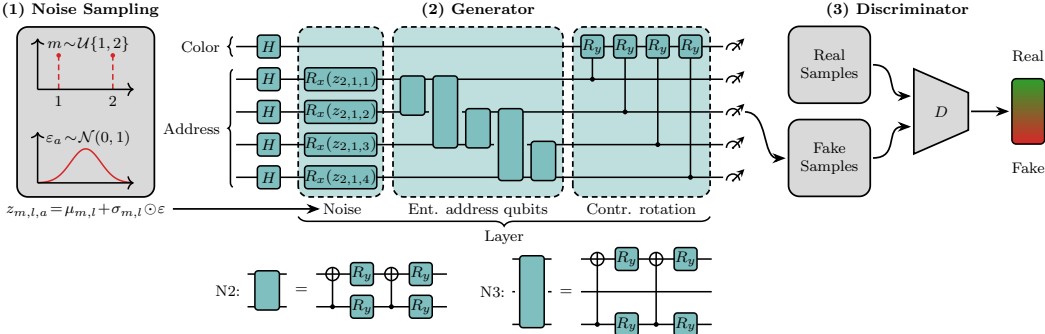

Figure 1: **Overview of the proposed QGAN generator and training workflow for a** $4 \times 4$-**pixel grayscale image.** (1) Noise Sampling: a multimodal latent distribution is formed by uniformly sampling a discrete mode index $m \in \{1, 2\}$ and drawing Gaussian noise $\varepsilon_a \sim \mathcal{N}(0, 1)$. The learnable affine transformation $z_{m,\ell,a} = \mu_{m,\ell,a} + \sigma_{m,\ell,a} \varepsilon_a$ produces tuned noise inputs. (2) Quantum Generator: the generator circuit begins with Hadamard gates preparing an equal superposition (gray image). Each layer consists of noise uploading via parametrized $R_x$ rotations, (entanglement across address qubits using alternating nearest-neighbor (N2) and next-nearest-neighbor (N3) two-qubit gates, and (controlled $R_y$ rotations on the color qubit to encode pixel intensities. The decompositions of the N2 and N3 gates into $R_y$ rotations and CNOTs are shown below. (3) Discriminator: the quantum state is decoded into an image and passed to a classical CNN critic $D$, whose scalar Wasserstein score provides gradients for training both generator and discriminator.

The state $|j\rangle$ of the $A$ so-called *address qubits* tracks the position index $j$ and the state $|c(\boldsymbol{x}_j)\rangle$ encodes the corresponding data value $\boldsymbol{x}_j$. For grayscale images, we use the *flexible representation of quantum images (FRQI)* (Le et al., 2011a;b). In this encoding, $\boldsymbol{x}_j$ is a scalar with the grayscale value of that pixel. We encode this information in the $z$-polarization of the color qubit

$$|c(x_j)\rangle = \cos(\tfrac{\pi}{2}x_j)|0\rangle + \sin(\tfrac{\pi}{2}x_j)|1\rangle, \tag{2}$$

with the pixel value normalized to $x_j \in [0, 1]$. Thus, combining Eqs. (1) and (2), a $2^A$-pixel image is encoded into a state with $n = A+1$ qubits. FRQI color extensions are discussed in App. B.2.3.

The order in which the pixels are indexed can change the entanglement entropy of the resulting state (Jobst et al., 2024). Here, we choose hierarchical indexing based on the so-called $Z$- or Morton order (Latorre, 2005; Le et al., 2011a;b; Jobst et al., 2024): the first two bits of the index $j$ label the quadrant of the image the pixel is in, the next two bits label the subquadrant, and so on. This tends to decrease the entanglement entropy compared to other orderings, resulting in more compressible states (see Jobst et al. (2024) for grayscale images and Kiwit et al. (2025) for color images).

## 3 METHOD

In GAN training, the generator $G_{\boldsymbol{\theta}}(\boldsymbol{z}) \mapsto \boldsymbol{x}$ aims to map a noise vector $\boldsymbol{z}$ to a sample $\boldsymbol{x}$, indistinguishable from real data, while the discriminator $D_{\phi}(\boldsymbol{x})$ aims to differentiate between real and fake samples. In our setup, the generator is a quantum circuit while the discriminator is a classical convolutional neural network. Both are trained jointly using the gradient-penalized Wasserstein GAN (Gulrajani et al., 2017) scheme. Wasserstein extensions of QGANs were previously applied to (patch-based) image generation (Tsang et al., 2023) and other classical tasks (Herr et al., 2021), while Chakrabarti et al. (2019); Kiani et al. (2022) introduced Wasserstein distances in QGANs earlier for quantum data. Here, we focus on our main methodological contribution: the design of the quantum generator $G$, which introduces an enhanced noise input and a circuit architecture tailored towards the generation of FRQI states. As illustrated in Fig. 1, the QGAN training passes through three stages: (1) *Noise Sampling*, (2) *Application-Specific Generator Design*, and (3) *Discriminator*. Quantum gate definitions are provided in App. A. Further implementation details and a discussion on the inductive bias of the circuit towards FRQI states are provided in App. B.

**(1) Noise sampling.** Real images exhibit strong statistical structure, with pixel intensities often forming multiple well-separated modes rather than a single unimodal distribution. For example, the

central pixel of the MNIST digits *0* and *1* shows a clear bimodal pattern reflecting the black and white intensities as depicted in Fig. 2 (right side). To reflect this intrinsic multimodality in the latent space, we introduce multimodal noise inputs combined with a learnable noise-tuning mechanism. Instead of injecting a fixed unimodal Gaussian, commonly used in prior QGANs (Riofrío et al., 2024; Tsang et al., 2023; Ma et al., 2025), we parameterize a Gaussian mixture whose component means and variances are learned jointly with the generator.

To generate an image $x$, we sample the noise vector from a multivariate isotropic Gaussian $\varepsilon \sim \mathcal{N}(0, I_A)$, where $A$ is the number of address qubits. The same noise vector is shared across all $L$ generator layers. We then sample the mode index from a discrete uniform distribution $m \sim \mathcal{U}\{1, \ldots, M\}$ and select the corresponding reparameterization matrices $\mu_m, \sigma_m \in \mathbb{R}^{L \times A}$ that are part of the learnable generator parameters $\theta$. Finally, we apply the element-wise affine transformation $z_{m,l} = \mu_{m,l} + \sigma_{m,l} \odot \varepsilon$ with $l \in \{1, \ldots, L\}$. Hence, the noise input becomes distinct for each mode $m$ and layer $l$. This results in sampling a (uniform) Gaussian mixture model $z \sim \frac{1}{M} \sum_{m=1}^{M} \mathcal{N}(z \mid \mu_m, \mathrm{diag}(\sigma_m^2))$, where $\mu_m, \sigma_m \in \mathbb{R}^{LA}$ correspond to the flattened matrices $\mu_m$ and $\sigma_m$. This noise tuning technique is inspired by the reparametrization trick (Kingma & Welling, 2014). For each single noise component with $a \in \{1, \ldots, A\}$, this can be represented by rotations on address qubit $a$ of the form

$$\boxed{R_x(z_{m,l,a})} \quad = \quad \boxed{R_x(\mu_{m,l,a} + \sigma_{m,l,a}\varepsilon_a)}. \tag{3}$$

In terms of its quantum circuit implementation, this tuning corresponds to $M$ rotation gates encoding unimodal noise components but controlled by *classical* bits encoding the sampled mode index $m$ to realize each mode via a separate controlled gate layer. Figure 2 presents the bimodal case.

**(2) Application-specific generator design.** The quantum generator employs a circuit ansatz with an inductive bias tailored towards the FRQI representation. Analogously for MCRQI, a color-extended task-specific ansatz is proposed in App. B.2.3. It starts with a layer of Hadamard gates to bring the initial state $|0\rangle^{\otimes(A+1)}$ into an equal superposition, which resembles a valid FRQI state of a uniformly gray image. After the Hadamard gates, (multiple) layers of the generator are added. Each layer consists of (a) noise uploading gates, (b) gates that entangle the the address qubits, and (c) controlled rotations of the color qubits, as depcited in Fig. 1 (middle) and described in the following.

(a) First, the sampled noise is injected by parameterized $R_x$ gates.

(b) Second, entangling gates are arranged as a ladder alternating between connecting nearest-neighbor (N2) and next-nearest-neighbor (N3) address qubits. Due to the Morton order, as described in Sec. 2, N2 gates mix qubits addressing two different spatial dimensions (vertical and horizontal). Consequently, N3 gates only mix between qubits addressing the same spatial dimension at different scales. We refer to repeating these ladders $\ell$ times as introducing $\ell$ *sub-layers*. Each sub-layer uses distinct parameters and alternates the direction of qubit connections between top-down and bottom-up. The address qubit pairs are entangled by parameterized gates defined in Fig. 1 (bottom). These gates realize compressed orthogonal two-qubit transformations, which have proven effective for encoding FRQI states (Kiwit et al., 2025).

(c) Third, we rotate the color qubit via parametrized $R_y$ gates, controlled by a single address qubit, which modulates the color of half the pixels in general FRQI images and leaves the other half unchanged. The pixels that are affected are those whose corresponding address bit is set to one in the binary representation of their index. Importantly, the preceding address qubit entangling gates can emulate *multi*-control color rotations, which simultaneously affect fewer pixels.

As a final step, the state generated by the quantum circuit must be decoded into an image. It is essential to note that the ansatz does not enforce valid FRQI states, i.e., neither nonnegative real amplitudes nor a uniform superposition over address qubits (uniform pixel distribution upon measurement) are guaranteed. Normalizing/conditioning the computational basis probabilities enables decoding as valid FRQI states via trigonometric inverse functions, as further detailed in App. B.2.4.

**(3) Discriminator.** The discriminator, illustrated in Fig. 1 (right), is a classical convolutional neural network that receives both real training images and decoded samples produced by the quantum generator. Real and fake inputs are processed through the same CNN, which outputs a scalar critic

Figure 2: Illustration of multimodal noise modeling (left to right). Quantum circuit perspective of implementing a bimodal mixture distribution via controlled rotations sampling the classical bit $m$ uniformly and $\varepsilon$ normally (unimodal). $z_0$ and $z_1$ denote the tuned noise (shifted by 0 and $\pi$, respectively). In this single-pixel example, noise is injected directly into the color qubit (no address qubits or layering), so layer and qubit indices $l, a$ as in Eq. (3) are omitted. The noise separates the prepared states around $|0\rangle$ and $|1\rangle$ in the Bloch sphere. Measurements yield pixel values via the probability of $|1\rangle$, consistent with FRQI states in Eq. (2). As an example, the distribution resembles the bimodal statistics of the MNIST center pixel for handwritten digits *0* and *1*, with peaks at *0* (black) and *1* (white) and vanishing probability in between, avoiding unrealistic gray pixels.

score in accordance with the Wasserstein GAN formulation. Higher scores correspond to more realistic images, while lower scores indicate generated samples, thereby providing the gradient signal used to update both the discriminator and the quantum generator.

To the best of our knowledge, multimodal latent distributions have only been considered implicitly in quantum *conditional* models (Liu et al., 2021; Zeng et al., 2023), and their explicit treatment is novel in QGANs, with only classical analogues reported (Gurumurthy et al., 2017). Furthermore, existing QGAN approaches (Riofrío et al., 2024; Tsang et al., 2023; Ma et al., 2025) rely solely on *noise re-uploading* (Pérez-Salinas et al., 2020) to enhance expressivity through complex, non-linear dependencies on the noise input, and do not incorporate the tuning technique we propose. Our tuned multimodal latent space therefore constitutes a novel inductive bias for QGANs, which is crucial for preventing the blending of artifacts and enabling rich intra-class variation, as demonstrated in Sec. 4.2. In addition, our generator circuit introduces a task-aligned ansatz tailored to the FRQI representation. This modular design constitutes a second, complementary inductive bias that differs fundamentally from prior QGAN architectures, which typically rely on generic circuits.

## 4 RESULTS

We designed the experiments with three main objectives: (i) demonstrating the high quality and diversity of the QGAN image generation, (ii) analyzing the impact of our QGAN design choices, and (iii) assessing the transferability to future quantum computers under inevitable shot noise in the generation process. All experiments are conducted in numerical simulation. We evaluate our approach using standard image datasets, including the grayscale MNIST (Lecun et al., 1998; Deng, 2012) and Fashion-MNIST (Xiao et al., 2017) datasets. These datasets contain ten classes of different handwritten digits and clothing photos, respectively. Both have a resolution of $28 \times 28$ pixels and are interpolated (bilinear) to $32 \times 32$ pixels to match 11-qubit FRQI states. The $32 \times 32$-pixel Street View House Numbers (SVHN) color images (Netzer et al., 2011) are represented by 13-qubit MCRQI states. Further details on the datasets are provided in App. C. All images presented are generated from QGANs trained for a fixed number of iterations or, when stated, loaded from a checkpoint that minimizes the maximum mean discrepancy (MMD; see App. D.2). For clarity, images are manually ordered and, where relevant, matched to classes. We vary and indicate the number of generator layers, but place two sub-layers each. Implementation details and configurations can be found in App. B. We use the Fréchet Inception Distance (FID) (Heusel et al., 2017) to quantify image quality and diversity. Lower FID is better, with zero for identical distributions. For reference, an untrained QGAN producing noisy gray images typically yields FIDs in the mid-hundreds. FID should not be compared across different datasets or subsets. See App. D.1 for details and limitations.

### 4.1 GENERATING SAMPLES OF HIGH QUALITY AND DIVERSITY

To demonstrate image generation of high quality and diversity, we train large QGAN models with 64 layers and 40 noise modes for about $50\,000$ generator updates on the full MNIST and Fashion-

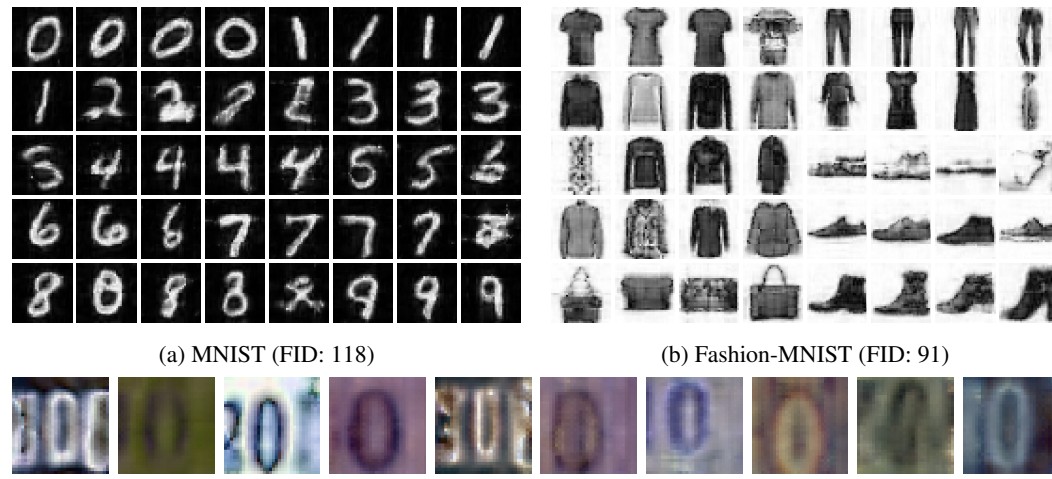

(a) MNIST (FID: 118)  (b) Fashion-MNIST (FID: 91)

(c) SVHN, class *0* (FID: 84)

Figure 3: QGAN samples for (a) MNIST, (b) Fashion-MNIST, and (c) SVHN. For (a) and (b), one image is shown for each of the 40 noise modes used by the large QGANs (64 layers). For each mode, the displayed image is selected as closest to the mean of 500 samples in Euclidean distance. For (c), a 32-layer QGAN generates images restricted to containing the digit *0*. The central digit is consistently a *0*, while extra digits may occur on the sides, reflecting typical house number tags.

MNIST datasets and present the checkpoint that minimizes the MMD metric. As shown in Fig. 3, not only are all ten classes successfully captured with high visual quality, but images also reveal rich intra-class diversity. The depth of the models enables them to represent fine image structures in digits (Fig. 3a), or extreme cases such as the single-pixel-wide straps in the *sandals* class (Fig. 3b), which demand more complex entanglement among the address qubits. The size of the quantum generator may appear large relative to previous works. However, since these works only covered small subsets of classes within these datasets, less expressive models suffice. Similarly, our QGAN framework also learns high-quality images with shallower circuits on these subsets. In App. D.3, we analyze this trade-off in more detail and observe that deeper models are necessary not only to improve image quality on a fixed dataset but also to maintain quality when scaling to all classes.

**A colorful extension.** The model is also trained on the color dataset, SVHN, restricted to images containing the digit *0*. In this setting, the *0* consistently occupies the central position, while additional digits may appear on the left and right. Consequently, the surrounding context introduces variability, as house numbers naturally contain multiple digits, and the background colors may also differ. Fig. 3c illustrates representative results from a QGAN model with 32 layers of the color-extended task-specific ansatz and 3 modes, trained for nearly $100\,000$ iterations and evaluated via MMD. One can observe that the central digit is reliably reconstructed as a *0*, while digits occurring to the left often resemble *2*s or *3*s, reflecting the realistic distribution present in the dataset.

### 4.2 IMPACT OF TASK-SPECIFIC GENERATOR DESIGN CHOICES

We analyze the impact of the two main design choices in the presented QGAN framework, concerning the generator design and noise techniques, through additional experiments. Beyond analyzing the final images, App. D.5 presents a layer-wise study of entanglement formation in the generator.

**Task-specific generator design ablation study.** We evaluate the relevance of two generator design choices specific to the task of image generation: (i) the generator circuit ansatz specific to the image state encoding instead of a task-agnostic ansatz (see App. B.2.1), and (ii) the FRQI state representation over simple amplitude image encoding. Compared to the layers in the task-specific ansatz, the task-agnostic ansatz implements entanglement via fixed cyclic N2 controlled-NOT gates, while parameterization occurs only in single-qubit $z - y - z$ rotation sequences. We perform an ablation study that compares the results of QGANs where these design choices are either implemented or

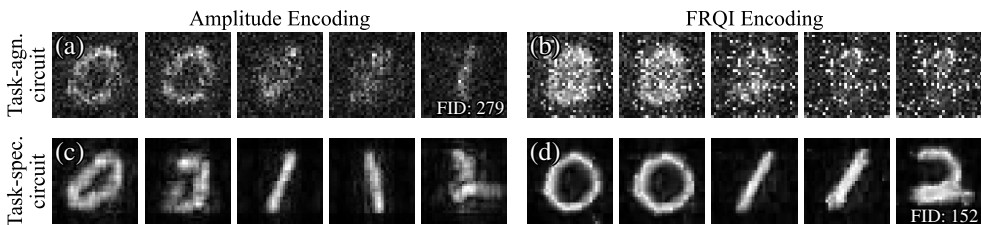

Figure 4: Ablation study highlighting the importance of task-specific model design choices. Panels (a) and (b) show images from the task-agnostic circuit using Amplitude encoding and FRQI encoding, respectively. Panels (c) and (d) show images from the task-specific circuit using Amplitude encoding and FRQI encoding, respectively. Task-specific modifications yield clearer, less distorted digit representations, with combining both proposed design choices leading to the best results (d).

omitted. All combinations use 16 layers, and are trained for $15\,000$ iterations on the digits *0*, *1* and *2*. Furthermore, the enhanced noise inputs (3 modes) may improve even the amplitude encoding and task-agnostic ansatz combination, which most closely resembles the setup by Tsang et al. (2023).

Figure 4 shows the results, revealing the impact of the two design choices. The most pronounced difference in image quality arises from the ansatz choice. The task-agnostic ansatz (Fig. 4a, 4b) produces images with a vague glimpse of digits. Furthermore, this ansatz produces images of limited diversity, particularly omitting classes, such as digit *2*. Formally, this corresponds to mode collapse, which limits QGANs with task-agnostic ansätze from scaling to more classes, as in previous works limited to at most three classes. The task-specific ansatz (Fig. 4c, 4d) clearly achieves what the task-agnostic one fails to model: spatial coherence and defined edges—two main properties of natural images (Simoncelli & Olshausen, 2001). Hence, neighboring pixels exhibit similar colors, with edges clearly defined rather than being fuzzy.

For the image encoding choice, the overall contrast of the digits from the black background is improved when transitioning from amplitude (Fig. 4a, 4c) to FRQI encoding (Fig. 4b, 4d). We observe that the saturation is more balanced across different samples and more uniform within each digit. These results support the theoretical expectation of sensitivity in saturation for amplitude-encoded images due to the need of amplitude normalization. FRQI encoding handles the saturation by introducing the color qubit. In addition, the edges are less blurred when switching from amplitude to FRQI encoding under the image-specific ansatz. We tested two image-specific ansatz realizations for amplitude encoding (Fig. 4c): one omits the layer of controlled color-qubit rotations, while the other replaces it with a layer of single-qubit rotations. No substantial differences were observed.

**From unimodal to multimodal noise through tuning.** In the following, we will discuss the role of input noise distributions and injection techniques, centered around generated images from three different experiments presented in Fig. 5. Given that previous QGAN works relied solely on unimodal noise distributions, we start the analysis with unimodal Gaussian noise (Fig. 5a). Pure blending by simply superimposing images of two classes (see *0*s where the inside of the circle is not transparent, e.g., leftmost image in Fig. 5a) is observed less frequently than in previous works (Tsang et al., 2023), which might be due to an improved generator design. However, more pronounced class mixing effects manifest as morphing shapes of distinct classes, such as *1*s appearing as right-leaning with curved tops and faint bottom bars reminiscent of *2*s (rightmost image in Fig. 5a). Although unimodal noise does not suffer from strict mode collapse onto a single digit, we conclude that scaling to datasets with many diverse classes is infeasible.

Introducing a multimodal distribution with three fixed modes (matching the number of classes included for training) mitigates these two mixing effects (Fig. 5b). However, this change is accompanied by a considerable loss in image quality, often obscuring visual class differentiation either (rightmost image in Fig. 5b). A likely reason is that sampling from modes placed at fixed $\mu_j$ away from zero results in noise injections that disrupt state preparation due to a systematic rotation in each layer, which the model can control only to a limited extent. Therefore, the proposed *noise tuning* technique, where the mode centers $\mu_j$ and widths $\sigma_j$ effectively become learnable parameters, is crucial for multimodality, generating clearly separated and undistorted images (Fig. 5c).

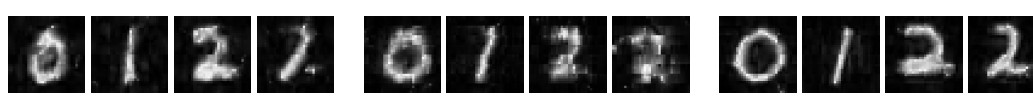

(a) Unimodal (FID: 216)    (b) Multi. w/o tuning (FID: 201)    (c) Multi. w/ tuning (FID: 152)

Figure 5: Comparison of noise inputs: (a) unimodal, (b) fixed multimodal, (c) tuned multimodal. Models were trained on MNIST classes *0–2*, with 3 modes in the multimodal setups. Images are generated after 15 000 training iterations and manually selected to highlight characteristic effects.

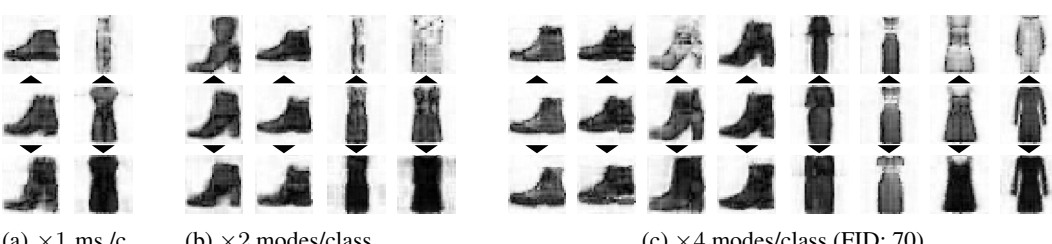

(a) ×1 ms./c. (FID: 103)    (b) ×2 modes/class (FID: 97)    (c) ×4 modes/class (FID: 70)

Figure 6: More input noise modes ("overmoding") diversify generated samples. Three models are trained on all ten Fashion-MNIST classes with a factor of (a) 1, (b) 2, and (c) 4 more noise modes than classes. We present all modes capturing the classes *ankle boot* and *dress*. Three images are shown per mode: the center image is closest to the mean in Euclidean distance, while the outer images closely approximate moves of $\pm 3\sigma$ along the first principal component (indicated by arrows). PCA, based on 1 000 samples per mode, illustrates the primary variability within each mode.

**More modes than classes ("overmoding").**    Choosing the number of modes equal to the number of classes is natural, however this information is unavailable in unsupervised datasets. Moreover, instances of the same class may exhibit very different features (high intra-class variety), and modeling them with more than a single mode might be an appropriate choice. By analogy to *overparameterization*, we call the use of an ansatz with a potential excess of modes *overmoding*. To analyze the effects of overmoding, we train three QGANs on the complete Fashion-MNIST datasets with ×1, ×2, and ×4 more modes than classes for 20 000 iterations (nearly 40 000 in the latter case). Fig. 6 shows generated images after training for the classes *ankle boot* and *dress*, corresponding to three models with 1, 2, and 4 modes per class.

Across all classes, increasing the number of modes enhances intra-class diversity by allowing the model to represent distinct sub-classes. A single mode (Fig. 6a) may already capture some variation, but typically sacrifices visual quality. In contrast, overmoding benefits both diversity and quality. With two modes (Fig. 6b), the model already separates flat vs. heeled boots and short vs. long dresses, which were previously conflated in a single mode. At four modes (Fig. 6c), the separation becomes more fine-grained. For *boots*, one heeled mode varies heel type (from stiletto, via block, to wedge), while another varies heel height. Flat-boot modes capture distinct styles, differing in details such as laces, soles, and pull tabs. *Dresses* are distinguished by sleeve type (long, short, cap, sleeveless/straps) and further vary in length within each mode. The fourth *dress* mode (Fig. 6c) transitions into the *coat* class by altering shape and introducing a zipper line. This overlap highlights the benefit of not conditioning QGAN modes on class labels, allowing the unsupervised model to exploit shared visual structures across classes. More *inter*-class modes are presented in App. D.4.

### 4.3 FINITE MEASUREMENT SHOT EFFECTS

The marginal distribution of the address qubits of valid FRQI states, after tracing out the color qubit, is uniform due to the sine–cosine structure in Eq. (2). In exact state-vector simulations without shot noise, the quality of the generated samples depends only on the ratio, rather than the absolute values, of the probability amplitudes of $|0\rangle$ and $|1\rangle$ in the color qubit for a given address. However, some basis states may have vanishingly small amplitudes in both $|0\rangle$ and $|1\rangle$. With a finite number of shots, such states are unlikely to be sampled, causing loss of pixel information or requiring unrealistic shot budgets. We show now that incorporating finite shot noise already during training alleviates this

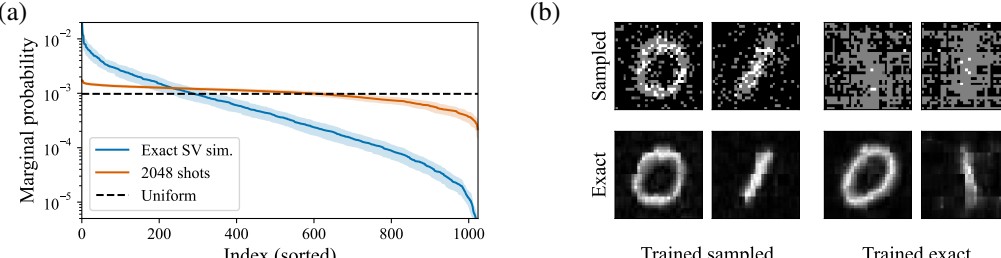

Figure 7: (a) Marginal probabilities of the address qubits sorted by magnitude. Exact state-vector simulations (blue) deviate strongly from the expected uniform distribution (dashed), with many amplitudes nearly zero, whereas finite sampling with 2 048 shots (orange) smooths the distribution toward uniformity. (b) Examples generated from 2 048 shots (top) and from exact probabilities (bottom). Finite-shot sampling introduces statistical noise that smooths the distribution and preserves pixel information. Hence, models trained on sampled data (left) yield clearer, more robust digits, while models trained on exact probabilities (right) tend to produce incomplete or distorted images.

problem. Very low probabilities may exclude information from some pixels, making it easier for the discriminator to detect fake samples and forcing the generator to avoid such cases and thus promoting more uniformly distributed marginal probabilities over the address qubits. Details of our implementation are presented in App. B.4.

Figure 7a illustrates how exact state-vector simulations yield highly uneven marginal probabilities across pixels, with many basis states exhibiting vanishingly small amplitudes. By contrast, sampling with a finite number of shots (2 048 in this example) smooths out the distribution and keeps the probabilities closer to the expected uniform distribution, thereby mitigating the risk of pixels being systematically excluded. This effect also shows in the sampled images in Fig. 7b, where finite shot noise ensures that pixel information is retained more consistently across the image. Together, these results highlight that incorporating shot noise into training not only prevents the discriminator from exploiting missing pixels but also promotes more robust and uniform sampling behavior. This uniformity is relevant for the scalability of our approach on real hardware under minimal total shot budgets $T$, i.e., a number of measurements proportional to the number of pixels $N$ suffices to decode all pixels to a sufficient precision, i.e., estimate each pixel with a variance $\mathcal{O}(N/T)$. In contrast, in non-uniform distributions, some marginal pixel probabilities may undesirably concentrate in the pixel count $N$. Generator gradient scaling with $N$, relevant for trainability, is analyzed in App. D.6.

## 5 DISCUSSION

In this work, we have made several contributions advancing quantum generative modeling. First, we demonstrated that end-to-end quantum Wasserstein GANs can be trained directly on full-resolution, standard classical image datasets without resorting to dimensionality reduction or patch-wise modeling, thus moving beyond the toy examples that have historically constrained the field. Second, we showed that performance depends critically on the incorporation of inductive biases through carefully designed circuit architectures, rather than relying on generic, application-agnostic ansätze, and multi-modal noise injections. Our findings highlight that task-specific architectural choices are not only a technical detail but a central driver of scalability and generative quality in quantum machine learning. Finally, by training under realistic shot-noise conditions, we provide a practical pathway toward robust quantum image generation, especially for early fault-tolerant quantum computing with a moderate number of logical qubits. Together, these contributions underscore that progress in quantum generative modeling will come not only from hardware advances but also from principled design choices that align quantum models with the structure of the task.

Our work aims to advance quantum image generation. However, we emphasize that neither outperforming classical approaches nor demonstrating quantum advantage is a direct goal. For quantitative comparisons with prior works, we employ the Fréchet Inception Distance (FID) (Heusel et al., 2017), a widely adopted metric that reflects both visual quality and diversity, albeit with known limitations discussed in App. D.1. As our primary quantum baseline, we compare against the patch-generation

QGAN of Tsang et al. (2023), with detailed benchmarking results in App. D.7. The comparison is conducted for the largest data subsets and patch-QGANs Tsang et al. (2023). On the 3-class MNIST, our QGAN achieves an FID of $152$, improving upon the patch-QGAN (FID $207$). On the 2-class Fashion-MNIST, our model reaches an FID of $60$, substantially outperforming the patch-QGAN result of $179$. Among QGAN approaches that do not rely on substantial classical post-processing, these results place our QGAN framework ahead of the previous quantum state-of-the-art.

We now contextualize our QGAN performance relative to analogous classical GAN frameworks using the FID metric. We refer to the large-scale benchmark study by Lucic et al. (2018), and specifically to the classical Wasserstein GANs with gradient penalty. Across 100 hyperparameter-search runs, *mean* FIDs of approximately $55$ for MNIST and $85$ for Fashion-MNIST are reported. In comparison, our QGAN can achieve FIDs of $109$ on MNIST and $70$ on Fashion-MNIST, which is higher for MNIST, yet surpasses the classical mean for (10-class) FashionMNIST. Two key differences should be stressed. First, the classical generator is a four-layer neural network with nearly 9 million parameters (architecture from Chen et al. (2016)), whereas even our largest QGAN generators use over 100 times fewer parameters. Second, we did not perform an extensive hyperparameter search. While GAN training is known to be hyperparameter-sensitive (Lucic et al., 2018), this sensitivity is not unique to the quantum setting. The FID improvements over the largest QGAN models (Fig. 3) with different settings in other experiments indicate room for gains via hyperparameter tuning. Still, our results show that even with default settings, the QGAN framework effectively learns the image distributions. We note that modern classical generative models, especially diffusion-based image generators, are far more mature than current quantum approaches, including ours (Po et al., 2024).

A common criticism of quantum generative modeling with amplitude-type encodings such as FRQI concerns the apparent measurement overhead. An image of $N$ pixels can be encoded using only $\mathcal{O}(\log(N))$ qubits. Recovering an $N$-pixel image indeed requires $\mathcal{O}(N)$ measurements shots, however, this cost is not exponential in the problem size, which in this case is the number of pixels $N$. Rather, it is exponential in the number of qubits $n$, since $n \in \mathcal{O}(\log(N))$ for FRQI. Consequently, the measurement cost does not introduce a detrimental exponential overhead relative to the classical problem size. This characteristic is intrinsic to all amplitude-type encodings and is therefore not specific to our approach. We neither claim nor imply any exponential quantum speedup arising from the encoding or measurement process. Instead, our focus is on exploring practical architectures and training methods within the near-term hardware regime with few (logical) qubits. Beyond classical data generation, quantum generative models can function as quantum data loaders Zoufal et al. (2019), providing structured quantum states directly to downstream quantum algorithms as a compact, trainable interface. No classical readout is required, enabling more efficient hybrid workflows.

For completeness, we propose the following three ideas for decoding strategies to enhance the practical measurement scaling. One could use compressed sensing (Donoho, 2006; Candes & Tao, 2006; Candes et al., 2006) as a post-processing step. This method would act entirely classically: missing pixel intensities, i.e., non-measured states, can often be reconstructed from partial information using structural priors on natural images (Candes & Wakin, 2008; Duarte & Eldar, 2011). Quantum compressed sensing (Gross et al., 2010) could pose an interesting alternative. Otherwise, one could perform measurements in Fourier space. By applying the Quantum Fourier Transform to the address qubits, as suggested in the original FRQI framework (Le et al., 2011a;b), one could probe the frequency domain rather than pixel space. Since low-frequency components dominate natural images, higher-frequency qubits should naturally decouple, effectively concentrating measurement probability on the relevant low-frequency subspace. Finally, one could use shadow tomography techniques that leverage recent advances tailored to tensor-network states (Akhtar et al., 2023; Bertoni et al., 2024). By exploiting the limited bond dimension characteristic of natural images (Jobst et al., 2024), such methods could also provide theoretical error guarantees. Pursuing these directions could recast measurement overhead from a perceived limitation into an opportunity for additional streamlining, further aligning quantum generative modeling with the structure of natural data.

As a final reflection, it is striking to observe the disparity in resources required by quantum versus classical generative models for the datasets studied here. Our quantum approach achieves competitive synthetic data generation with only 11–13 qubits and on the order of ten thousand trainable parameters, whereas classical models typically rely on ten thousands of bits and hundreds of thousands and, typically, millions of parameters. This contrast highlights the remarkable expressive power that quantum computing can bring to machine learning, and we view it as yet another indication of its potential to fundamentally reshape how generative modeling is conceived and implemented.

## REPRODUCIBILITY STATEMENT

All experiments in this paper can be reproduced with the provided code and instructions. The complete codebase, which includes training scripts, evaluation notebooks, and configuration files, is included as part of the supplementary material for the submission. If the paper is accepted, we will make the code publicly available as a GitHub repository.

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

## A NOTATION AND DEFINITIONS

The present work follows the standard notions and definitions commonly found in the quantum computing literature (Nielsen & Chuang, 2011) and is briefly presented here.

We adopt the Dirac (*bra–ket*) notation, where a quantum state labeled by $\psi$ is written as a 'ket' $|\psi\rangle$. For a single qubit, $|\psi\rangle$ may be state zero $|0\rangle$, one $|1\rangle$, or, unlike a classical bit, in a *superposition*

$$|\psi\rangle = \alpha\,|0\rangle + \beta\,|1\rangle \quad \text{with} \quad \alpha, \beta \in \mathbb{C}, \quad |\alpha|^2 + |\beta|^2 = 1. \tag{4}$$

The coefficients $\alpha, \beta$ are called *probability amplitudes* for reasons that become clear shortly. The state-vector representation expresses the 'ket' states as column vectors when fixing a basis. The common *computational basis*, used in this work, is composed of the zero and one states as

$$|0\rangle = \boldsymbol{e}^{(0)} = \begin{pmatrix}1 & 0\end{pmatrix}^\top, \qquad |1\rangle = \boldsymbol{e}^{(1)} = \begin{pmatrix}0 & 1\end{pmatrix}^\top, \tag{5}$$

which span the *state* space is $\mathbb{C}^2$ and superpositions are simply basis decompositions. Equipped with the canonical inner product $\langle\phi|\psi\rangle$, this space is a Hilbert space. This definition uses a 'bra', which is the adjoint of the ket $|\psi\rangle$ (conjugate row vector of the state-vector), i.e., $\langle\psi| = |\psi\rangle^\dagger = (|\psi\rangle^\top)^*$.

The *tensor product* $\otimes$ combines single-qubit spaces into the joint state space $\mathbb{C}^{2^n}$ of an $n$-qubit system. For example, two qubits $|\psi_1\rangle$ and $|\psi_2\rangle$ form the composite state $|\psi\rangle = |\psi_1\rangle \otimes |\psi_2\rangle$, The computational basis naturally generalizes to $2^n$ states, given by all tensor products of $n$ qubits in $|0\rangle$ and $|1\rangle$, commonly labeled by a bit string or integer label, e.g., $|101\rangle \equiv |5\rangle$. Hence, the $n$-qubit Hilbert space is spanned by $\{|0\rangle, \ldots, |2^n - 1\rangle\} = \{\boldsymbol{e}^{(0)}, \ldots, \boldsymbol{e}^{(2^n - 1)}\}$.

*Entanglement* distinguishes two types of multi-qubit states. A state $|\psi\rangle \in \mathbb{C}^{2^n}$ is *separable* (unentangled) if a tensor product decomposition into single-qubit states exists $|\psi\rangle = |\psi_1\rangle \otimes \cdots \otimes |\psi_n\rangle$, and *entangled* otherwise. Hence, entangled states cannot be fully described by their subsystems, only by the joint system. In the FRQI representation used here, entanglement corresponds to spatially correlated pixel colors, whereas unentangled states yield pixel colors independent of position.

Quantum states evolve not only linearly $|\psi\rangle \mapsto U\,|\psi\rangle$, but also, which conserves normalization, by a unitary transformation, i..e, $U^\dagger U = UU^\dagger = I$. In the state-vector expression, this action corresponds to a matrix-vector product in a fixed basis. A standard way to express such transformations is through quantum circuits, where unitary operations are decomposed into elementary quantum gates (e.g., see Fig. 1). Two gates combine either sequentially, $U_1 \circ U_2$, corresponding in matrix form to $U_2 U_1$, or in parallel on disjoint subsystems via the tensor/Kronecker product $U_1 \otimes U_2$. The basic single-qubit gates used here are defined in the computational basis as

$$H = \frac{1}{\sqrt{2}}\begin{pmatrix}1 & 1\\ 1 & -1\end{pmatrix}, \quad X = \begin{pmatrix}0 & 1\\ 1 & 0\end{pmatrix}, \quad Y = \begin{pmatrix}0 & -i\\ i & 0\end{pmatrix}, \quad Z = \begin{pmatrix}1 & 0\\ 0 & -1\end{pmatrix}. \tag{6}$$

Parameterized rotation gates are generated by the Pauli operators $X, Y, Z$ through exponentials

$$R_x(\theta) = e^{-i\theta X/2}, \quad R_y(\theta) = e^{-i\theta Y/2}, \quad R_z(\theta) = e^{-i\theta Z/2}, \tag{7}$$

rotating a qubit about its $x$-, $y$-, and $z$-axis by an angle $\theta$, respectively. Controlled (two-qubit) gates act conditionally, with the control qubit determining whether the operation is applied to the target qubit. Examples include the controlled-NOT (CNOT) and controlled-$R_y$ (in block-matrix notation):

$$\text{CNOT} = \begin{pmatrix}\boldsymbol{I}_2 & 0\\ 0 & X\end{pmatrix}, \quad cR_y(\theta) = \begin{pmatrix}\boldsymbol{I}_2 & 0\\ 0 & R_y(\theta)\end{pmatrix}. \tag{8}$$

Note that only multi-qubit gates can alter the entanglement of a state.

Finally, the probabilistic nature of quantum mechanics arises from the fact that quantum states cannot be fully observed: measurements yield probabilistic outcomes and collapse the state to align with the observation. For computational basis measurements, the probability of observing the qubits representing integer $i \in \{0, \ldots, 2^n - 1\}$ is

$$p_i = |\langle i|\psi\rangle|^2. \tag{9}$$

This probability is the squared magnitude of the corresponding probability amplitude in the superposition of computational basis states (or, put differently, the inner product of the $|\psi\rangle$ and $|i\rangle$). Consequently, the closer $|\psi\rangle$ is to a basis state $|i\rangle$, the higher the likelihood of observing $i$ upon measurement. In a quantum computer, states can typically be prepared repetitively. Therefore, from a number of measurement shots, certain state quantities, such as the (computational basis) probabilities, can be estimated, which are of particular interest to decode the image from an FRQI state.

## B  METHODOLOGICAL AND IMPLEMENTATION DETAILS

All experiments in this work are implemented as numerical state-vector simulations. For the gradient-based optimization, we use PennyLane (Bergholm et al., 2022) in combination with the just-in-time compilation and vectorization capabilities of JAX (Bradbury et al., 2025) to perform auto-differentiable, GPU-accelerated state-vector calculations.

### B.1  GENERATIVE MODELING

The Generative Adversarial Networks (GAN) (Goodfellow et al., 2014) technique was originally proposed for classical neural networks. One neural network functions as the *generator* $G_{\boldsymbol{\theta}}(\boldsymbol{z})$ and learns (parameters $\boldsymbol{\theta}$) to produce samples, based on random noise inputs $\boldsymbol{z}$, that are indistinguishable from the real data. In contrast, another neural network operates as the *discriminator* $D_{\boldsymbol{\phi}}(\boldsymbol{x})$ and concurrently learns (parameters $\boldsymbol{\phi}$) to provide a discrimination signal indicating whether the input is real or generated (fake). GANs can be readily extended to quantum generative models by replacing the generator neural network with a generator quantum circuit, where the generated data sample is constructed from measurement expectation values for continuous-valued outputs (Riofrío et al., 2024), such as images (Tsang et al., 2023). In principle, although not studied in this work, the discriminator could also be a quantum model.

GANs were originally introduced with a discriminator resembling a binary discrimination signal (for classification $D_{\boldsymbol{\phi}}(\boldsymbol{x}) = 1$ for real and $D_{\boldsymbol{\phi}}(\boldsymbol{x}) = 0$ for fake inputs $\boldsymbol{x}$). Due to training instability and problems such as the mode collapse phenomenon (resulting in less diverse samples than the real distribution), the original GAN framework can be improved by the Wasserstein-GAN (WGAN) approach (Arjovsky et al., 2017), where the discriminator now provides a continuous discrimination signal $D_{\boldsymbol{\phi}}(\boldsymbol{x}) \in \mathbb{R}$ that should be maximized for real and minimized for fake inputs $\boldsymbol{x}$. This is described by the following optimization problem, which directly gives rise to the corresponding loss functions that are minimized alternately during training:

$$\min_{\boldsymbol{\theta}} \max_{\boldsymbol{\phi}} \underbrace{\overbrace{\mathbb{E}_{\boldsymbol{x} \sim \mathbb{P}_{\boldsymbol{x}}} D_{\boldsymbol{\phi}}(\boldsymbol{x})}^{=\mathcal{L}_G(\boldsymbol{\theta})} - \mathbb{E}_{\boldsymbol{z} \sim \mathbb{P}_{\boldsymbol{z}}} D_{\boldsymbol{\phi}}(G_{\boldsymbol{\theta}}(\boldsymbol{z}))}_{=-\mathcal{L}_D(\boldsymbol{\phi})} \tag{10}$$

The noise distribution $\mathbb{P}_{\boldsymbol{z}}$ induces the generation distribution $\mathbb{P}_{G_{\boldsymbol{\theta}}}$ through the map from noise to data space that the generator $G_{\boldsymbol{\theta}}(\cdot)$ provides. We utilize batches of size $B$ of generated (and real) data to evaluate the *empirical* loss functions $L_G(\boldsymbol{\theta})$ and $L_D(\boldsymbol{\theta})$, which estimate the expectations over the noise and real data distributions $\mathbb{P}_{\boldsymbol{z}}, \mathbb{P}_{\boldsymbol{x}}$ in $\mathcal{L}_G(\boldsymbol{\theta})$ and $\mathcal{L}_D(\boldsymbol{\theta})$, respectively, by substituting $\mathbb{E}(\cdot) \approx \frac{1}{B} \sum(\cdot)$.

The discriminator is required to be 1-Lipschitz so that its output differences reflect actual distances in input space, preventing it from creating artificial in the loss landscape that would distort the Wasserstein distance. To enforce this condition, it is common practice to add a gradient penalty of the discriminator with respect to its inputs, scaled by a regularization coefficient $\lambda > 0$

$$\mathcal{L}_D(\boldsymbol{\phi}) \leftarrow \mathcal{L}_D(\boldsymbol{\phi}) + \lambda \mathbb{E}_{\hat{\boldsymbol{x}} \sim \mathbb{P}_{\hat{\boldsymbol{x}}}} \left[ (\|\nabla_{\hat{\boldsymbol{x}}} D(\hat{\boldsymbol{x}})\|_2 - 1)^2 \right], \tag{11}$$

where these inputs $\hat{\boldsymbol{x}}$ are uniformly distributed $\mathbb{P}_{\hat{\boldsymbol{x}}}$ on lines between pairs of samples from the data distribution $\mathbb{P}_{\boldsymbol{x}}$ and generator distribution $\mathbb{P}_{G_{\boldsymbol{\theta}}}$ (Gulrajani et al., 2017). Again, finite batches of $B$ inputs provide expectation estimates and yield the gradient-penalty version of the empirical loss $L_D(\boldsymbol{\phi})$. In this work, all implementations refer to the Wasserstein GAN method with gradient penalty (WGAN-GP), utilizing a quantum generator, whether it is termed QGAN or QWGAN.

### B.2  GENERATOR DESIGN

Beginning with precisely defining the task-agnostic circuit used for the quantum generator baseline, we provide further theoretical arguments on the proposed task-specific circuit design with respect to encoding inductive bias with respect to typical families of FRQI image transformations. Finally, the extension to color images is specified, and the decoding procedure for invalid FRQI states is detailed.

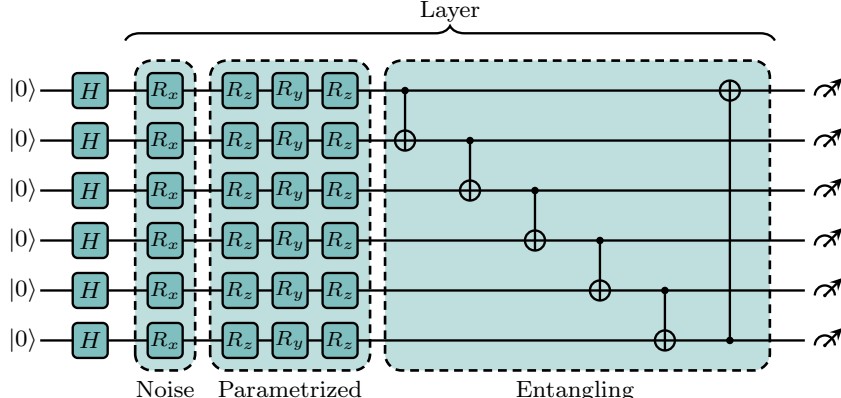

Figure 8: Task-agnostic generator circuit ansatz used in the ablation study. Each layer consists of (i) a noise injection block implemented by single-qubit $R_x$ rotations acting on all qubits, (ii) a parameterized block of local $R_z R_y R_z$ rotations, and (iii) a fixed cyclic entangling block of CNOT gates. All qubits are initialized in the $|+\rangle$ state via Hadamard gates, and the same layer structure is repeated throughout the circuit. This architecture is independent of the underlying image encoding and does not exploit any specific structure of the target data.

### B.2.1 TASK-AGNOSTIC CIRCUIT

As a baseline, we consider a *task-agnostic* generator ansatz, shown in Fig. 8, that is similar to circuits used in previous QGAN studies Tsang et al. (2023). Each layer first applies single-qubit $R_x$ rotations that inject classical noise into all qubits, followed by a block of parameterized $R_z R_y R_z$ rotations on each qubit and a fixed cyclic pattern of CNOT gates implementing entanglement across the register. A schematic illustration is given in Fig. 8. All qubits are initialized in the $|+\rangle$ state, and the same layer structure is repeated without any dependence on the chosen image encoding or the spatial layout of pixels. This architecture therefore serves as a generic, hardware-efficient variational circuit with linear parameter scaling in the number of qubits, but it lacks the inductive bias needed to explicitly encode local spatial correlations or separate address and color degrees of freedom, in contrast to our task-specific ansatz.

### B.2.2 INDUCTIVE BIAS OF THE TASK-SEPCIFIC CIRCUIT

The FRQI formalism supports a structured set of unitary image transformations that operate on color information, positional information, or their combination. As shown in Fig. 9a-9c, these transformations fall into three categories (Le et al., 2011a). The operator depicted in Fig. 9a modifies only the color qubit by applying a single qubit unitary $U_1$ on the color qubit, producing uniform color adjustments across the entire image (such as global brightness or contrast shifts). The operator depicted in Fig. 9b introduces position-selective processing: a unitary $U_2$ acts on the color qubit, but only when the position register matches a specified computational-basis state or a set of states, enabling localized editing at specific pixel locations. Finally, The operator depicted in Fig. 9c act on the entire position register through an $n$-qubit unitary $U_3$, coupling spatial structure with color information. Prominent examples include the quantum Fourier transform, which redistributes color amplitudes according to spatial frequencies and can reveal structural features such as edges or periodic patterns in the transformed image. Together, these three operator classes provide a flexible toolbox for quantum image manipulation within the FRQI representation.

In the context of these FRQI image transformations, our task-specific generator architecture naturally inherits the expressive structure required to approximate all three operator classes. First, the controlled–$R_y$ rotations acting on the color qubit directly correspond to $U_1$-type transformations, implementing global intensity changes via single-qubit unitaries on the color register. Second, because each address qubit controls a subset of pixels in the Morton ordering, the generator's many controlled color rotations implement $U_2$-type localized transformations, where conditioning on individual address bits or their entangled superpositions allows selective modification of spatially

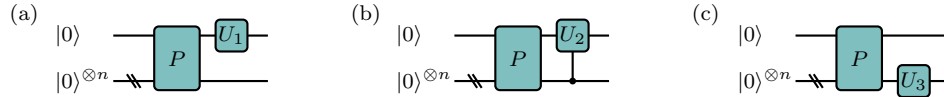

Figure 9: Overview of the three classes of FRQI image transformations as introduced by Le et al. (2011a). Each transformation acts on an FRQI state encoded by $P$, modifying either the color qubit, selected positions, or the joint color-position structure. (a) The single qubit unitary $U_1$ is applied to the color qubit and leave the position register unchanged (e.g. global color shifts). (b) A (multi)controlled unitary $U_2$ is applied to the color qubit conditioned on a subset of position basis states, enabling local modifications at specific pixel locations. (c) The operator acts on the position register via an $n$-qubit unitary $U_3$, thereby mixing spatial information with the color amplitudes as in quantum Fourier, wavelet, or cosine transform

coherent pixel groups. Finally, the alternating ladder of nearest- and next-nearest-neighbor entangling gates between address qubits emulates $U_3$-type positional transformations: these multi-qubit unitaries mix information across spatial scales, analogous to FRQI's position-space transforms such as the quantum Fourier transform. Thus, the generator circuit's structure, noise uploads, hierarchical address-qubit entanglement, and multi-controlled color rotations form an inductive bias that mirrors exactly the algebra of valid FRQI transformations. This explains why the model can efficiently represent realistic image transformations and why scaling to full-resolution datasets becomes feasible without dimensionality-reduction "tricks".

### B.2.3 Quantum Generative Modeling of Color Images

To extend the QGAN framework in this work to generating color images, we first present the extension of the FRQI grayscale encoding to color images proposed by Sun et al. (2011; 2013). Then, we introduce a natural extension of the task-specific, FRQI-based generator ansatz to this more general image encoding. We refer to this new ansatz as the *color-extended task-specific ansatz*.

**Quantum image representations for color images.** We encode color images with the *multi-channel representation of quantum images (MCRQI)* (Sun et al., 2011; 2013). For each pixel, the data value now has several components, $\boldsymbol{x}_j = (\begin{array}{cccc} x_j^R, & x_j^G, & x_j^B, & x_j^\alpha \end{array})^\top$, corresponding to the three RGB color channels and a possible fourth $\alpha$ channel indicating the opacity of the image. If only the three RGB channels are available for a given image (as is the case for all color image datasets considered in this work), the image is at full opacity and we can simply set the $\alpha$ channel to zero (Sun et al., 2013) or ignore it in the decoding. The color information of a pixel is then encoded in a three-qubit state as

$$
\begin{aligned}
|c(\boldsymbol{x}_j)\rangle = \frac{1}{2}s \Big( & \cos(\tfrac{\pi}{2}x_j^R)\,|000\rangle + \sin(\tfrac{\pi}{2}x_j^R)\,|100\rangle \\
& + \cos(\tfrac{\pi}{2}x_j^G)\,|001\rangle + \sin(\tfrac{\pi}{2}x_j^G)\,|101\rangle \\
& + \cos(\tfrac{\pi}{2}x_j^B)\,|010\rangle + \sin(\tfrac{\pi}{2}x_j^B)\,|110\rangle \\
& + \cos(\tfrac{\pi}{2}x_j^\alpha)\,|011\rangle + \sin(\tfrac{\pi}{2}x_j^\alpha)\,|111\rangle \Big),
\end{aligned}
\tag{12}
$$

with normalized values $x_j^R, x_j^G, x_j^B, x_j^\alpha \in [0,1]$. Thus, by inserting this definition in Eq. (1), a color image with $2^A$ pixels, where $A$ is the number of address qubits, is encoded into a quantum state with $n = A + 3$ qubits. Just as for grayscale images, encoding natural color images via MCRQI results in lowly-entangled states, which are well approximated by tensor-network states. To prepare the state *exactly* on a quantum computer, we can essentially reuse the same circuit that prepares an FRQI state and run it for each color channel separately. However, this procedure treats the color channels independently, which is not ideal for generative modeling, as channels should be considered together.

**Color-extended task-specific ansatz.** In MCRQI, the three color qubits, as defined in Eq. (12), play distinct roles: the first (left) encodes the channel intensity in its $z$-polarization, while the last two (center and right) specify the channel, i.e., $|00\rangle$ for R, $|01\rangle$ for G, $|10\rangle$ for B, and $|11\rangle$ for $\alpha$.

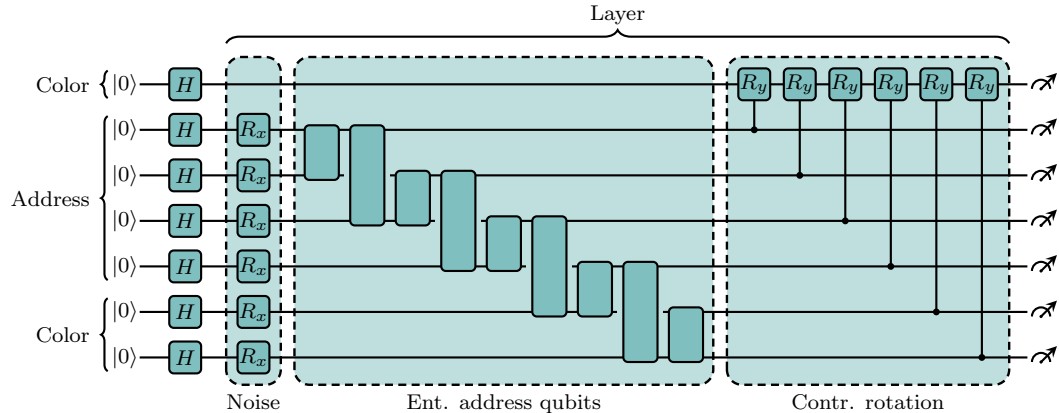

Figure 10: Quantum generator for a $4 \times 4$-pixel color image with one layer of noise, entangling and controlled $R_y$ gates. The last two color qubits are interpreted as (channel-) address qubits, analogous to four sub-pixels per pixel, and are integrated into the address qubit register accordingly.

Interpreting these two channel qubits also as address qubits casts MCRQI into an FRQI perspective, effectively mapping a color image onto a grayscale image of doubled resolution. Within the Morton order, when these channel addressing color qubits are placed as the last two address qubits, this can be interpreted as subdividing each pixel into four sub-pixels. This interpretation aligns with the design of digital displays, where each pixel is divided into RGB sub-pixels that, when sufficiently miniaturized, appear as a single colored pixel to the human eye. We adopt this physical intuition as the basis for our color-extended ansatz to achieve a task-specific design with sufficient inductive bias.

Consequently, extending our grayscale generator ansatz to color images becomes straightforward: the last two color qubits are treated as highest-resolution address qubits. Figure 10 provides a circuit diagram for a $4 \times 4$-pixel color image analogous to the $4 \times 4$-pixel grayscale example in Fig. 1. As with any address qubit, these two color qubits are affected by noise (the noise vector now includes two more components), are included in the N2 and N3 entangling ladders, and act as control qubits each for two additional $R_y$ gates on the (first) color qubit.

### B.2.4 GENERATOR DECODING: FROM QUANTUM STATES TO IMAGES

As outlined in the main text, the generator ansatz does not enforce valid FRQI states. Therefore, we construct the image solely from the estimated (computational basis) measurement probabilities of the generated state $|G(\boldsymbol{z}; \boldsymbol{\theta})\rangle$, and then normalize/condition these probabilities to recover a valid FRQI representation.

Concretely, for a pixel indexed by $j$, probabilities of observing the color qubit of pixel $j$ in states $|0\rangle$ and $|1\rangle$ are

$$p_{0,j} = \left|(\langle 0| \otimes \langle j|)|G(\boldsymbol{z}; \boldsymbol{\theta})\rangle\right|^2, \qquad p_{1,j} = \left|(\langle 1| \otimes \langle j|)|G(\boldsymbol{z}; \boldsymbol{\theta})\rangle\right|^2, \tag{13}$$

respectively, following the computational basis measurement definition in Eq. (9). The total probability of measuring information of pixel $j$ is

$$p_j = p_{0,j} + p_{1,j}. \tag{14}$$

For a valid FRQI state, the total probability always equals $1/2^A$ because all $2^A$ pixels are equally likely to be observed.

Hence, to achieve conformity to the FRQI representation in the decoding process, normalization uncovers the effective color-qubit amplitudes as defined in Eq. (2)

$$a_{0,j} = \sqrt{p_{0,j}/p_j}, \qquad a_{1,j} = \sqrt{p_{1,j}/p_j}. \tag{15}$$

Finally, the pixel value is derived from the FRQI encoding using trigonometric inverse functions as

$$x_j = \tfrac{2}{\pi} \arccos(a_{0,j}) = \tfrac{2}{\pi} \arcsin(a_{1,j}). \tag{16}$$

### B.3 Discriminator design

The discriminator is implemented as a convolutional neural network (CNN) (Lecun et al., 1998; Fukushima, 1980) designed to distinguish between real and fake images, i.e., those obtained by decoding the quantum states generated by the QGAN. The exact CNN architecture is adopted from the discriminator suggested by Gulrajani et al. (2017) for the MNIST dataset and outlined in the following. Three convolutional layers are used and followed by leaky ReLU activations, which preserve gradient flow in low-activation regions. All convolutions have $5 \times 5$ kernels and are applied with a stride of 2, which halves the size in each layer (no pooling is used). The number of convolutional filters is 64, 128, and 256 in the first, second, and third layers, respectively. After the convolutional layers, the outputs are flattened and passed into a fully connected layer that maps the extracted features to a single scalar output without any further activation function.

### B.4 Training with shot noise

We recall, that the exact probability distribution $P$ is defined as the squared amplitudes of the quantum state produced by the circuit. However, in practice, we only have access to samples from this distribution. Given the unfavorable scaling of the parameter-shift rule (Mitarai et al., 2018; Schuld et al., 2019) for large quantum systems, we focus on assessing the influence of shot noise on the generated distribution, but not the exact impact on the gradient. We define the computational basis $\{|x\rangle\}_{x \in \{0,1\}^n}$, i.e., the set of all bitstrings of length $n$, where $n$ is the number of qubits. The exact distribution $P$ assigns to each basis state $|x\rangle$ the probability $|\langle x|\psi\rangle|^2$, obtained from the squared amplitudes of the circuit's output state $|\psi\rangle$. In practice, however, we only have access to a finite-shot approximation $\hat{P}$, obtained from measurement samples. To emulate the effect of shot noise while keeping gradients tractable, we compute the per–basis-state deviation $\varepsilon(x) = \hat{P}(x) - P(x)$. We then perturb the exact distribution by this deviation, $\tilde{P}(x) = P(x) + \varepsilon(x)$, and apply a subsequent clipping step to ensure nonnegativity, followed by a renormalization. The gradient flows only through the exact distribution $P$, not through the stochastic deviation $\varepsilon$. This procedure closely resembles the reparameterization trick from (Kingma & Welling, 2014). Thus, the gradient is evaluated with respect to the exact distribution $P$, while still enabling efficient backpropagation during the simulation of quantum circuits. Note that the gradient is affected solely by the clipping step. If no measurement outcomes occur in the basis states corresponding both to $|0\rangle$ and $|1\rangle$, we assign the pixel a neutral gray value before reconstructing the image from the quantum state and feeding it into the discriminator.

### B.5 Training hyperparameters

Table 1 provides a comprehensive summary of all generator circuit configurations used in the experiments, along with estimates of the associated quantum resources. The minibatch size is 64 in most experiments, reduced to 32 for the large (64-layer) MNIST and Fashion-MNIST QGANs and to 16 for the color model, solely due to GPU memory limits. General generator parameters are initialized from a zero-centered normal distribution with variances $\sigma_{\text{init}}^2 \in \{0.001, 0.01, 0.025, 0.05\}$, using larger variances for smaller models and vice versa. Noise-tuning parameters are further scaled down by a factor of 10. The discriminator is updated ten times per generator update (all iteration counts in the paper refer to generator updates), with the ratio reduced to $5:1$ for the color model. Both the generator and discriminator are optimized with the *Adam* optimizer Kingma & Ba (2014), using learning rates in $\{0.001, 0.0025, 0.01\}$, typically lower for larger models. For the discriminator, the learning rate is reduced by a factor of 10 in grayscale experiments and 4 in the color model. Training the QGANs largely follows the WGAN-GP setup of Gulrajani et al. (2017), which informs the following choices: Adam hyperparameters are fixed to $\beta_1 = 0.5$ and $\beta_2 = 0.9$, and the gradient-penalty coefficient $\lambda$ is set to 10 as defined in Eq. (11).

## C Datasets

The MNIST dataset (Lecun et al., 1998; Deng, 2012) is a simple and widely used dataset for training machine learning models. It contains grayscale images of handwritten digits between '0' and '9', and associated labels indicating the correct digit. The original images have $28 \times 28$ pixels. Here, we

| Image shape | Color | Layers | Modes | Parameters | Qubits | CNOT est. | Depth est. |
|---|---|---|---|---|---|---|---|
| $4 \times 4$ | Gray | 8 (2) | 2 | 552 | 5 | 224 / 719 | 411 |
| $8 \times 8$ | Gray | 8 (2) | 2 | 888 | 7 | 384 / 1221 | 675 |
| $16 \times 16$ | Gray | 8 (2) | 2 | 1224 | 9 | 544 / 1723 | 939 |
| $32 \times 32$ | Gray | 8 (2) | 2 | 1560 | 11 | 704 / 2225 | 1203 |
| $32 \times 32$ | Gray | 8 (2) | 10 | 2840 | 11 | 704 / 2225 | 1203 |
| $32 \times 32$ | Gray | 16 (2) | 3 | 3440 | 11 | 1408 / 4417 | 2371 |
| $32 \times 32$ | Gray | 16 (2) | 10 | 5680 | 11 | 1408 / 4417 | 2371 |
| $32 \times 32$ | Gray | 32 (2) | 10 | 11360 | 11 | 2816 / 8801 | 4707 |
| $32 \times 32$ | Gray | 32 (2) | 20 | 17760 | 11 | 2816 / 8801 | 4707 |
| $32 \times 32$ | Gray | 32 (2) | 40 | 30560 | 11 | 2816 / 8801 | 4707 |
| $32 \times 32$ | Gray | 64 (2) | 40 | 61120 | 11 | 5632 / 17569 | 9379 |
| $32 \times 32$ | RGB | 32 (2) | 3 | 6944 | 13 | 3456 / 10791 | 5739 |
| $64 \times 64$ | RGB | 64 (2) | 10 | 27584 | 15 | 8192 / 25517 | 13491 |
| $128 \times 128$ | RGB | 64 (2) | 10 | 32320 | 17 | 9472 / 29491 | 15547 |
| $1024 \times 1024$ | RGB | 64 (2) | 10 | 46528 | 23 | 13312 / 41413 | 21715 |

Table 1: Different configurations for the quantum generator, using the proposed task-specific ansatz, and the corresponding requirements, including resource estimates. The number of sublayers is shown in parentheses alongside the number of layers. For quantum resource estimates (est.) via PennyLane (Bergholm et al., 2022), circuits are compiled over a gate set of single-qubit Pauli rotations and two-qubit CNOT gates, without extensive circuit optimization. The CNOT count is reported as the fraction of CNOT gates relative to the total number of compiled gates. The depth estimate corresponds to the maximum number of non-parallelizable gates in the compiled circuit. The bottom three configurations are hypothetical, i.e., beyond experimental studies in this work.

use bilinear interpolation to resize them to $32 \times 32$ pixels making them suitable for processing on a quantum computer. The class distribution over the $70\,000$ images is approximately uniform, with each class representing between $9\%$ and $11\%$ of the dataset.

The Fashion-MNIST dataset (Xiao et al., 2017) was introduced as a more challenging alternative to MNIST, after it became apparent that MNIST was too easily solved and no longer posed a significant challenge for more sophisticated classification models. The dataset also features $70\,000$ grayscale images with an original resolution of $28 \times 28$ pixels, which we again resize to $32 \times 32$ pixels using bilinear interpolation. Instead of handwritten digits, the images feature the 10 different clothing articles, T-shirt/top, trouser, pullover, dress, coat, sandal, shirt, sneaker, bag, ankle boot. The dataset is balanced over the ten classes. When presented in this work, the colors of the generated images are inverted for Fashion-MNIST for a more intuitive presentation, e.g., of shadings.

The Street View House Numbers (SVHN) dataset (Netzer et al., 2011) offers a natural-image analog to MNIST, comprising RGB $32 \times 32$ crops of digits 0–9 taken from Google Street-View scenes that feature real-world background variation. In our experiments, we restrict the corpus to those samples whose central digit is $0$. Within the official core split this subset contains roughly $4\,948$ training samples and $1\,744$ test samples.

# D  EXTENDED EXPERIMENTS AND ANALYSIS

Experiments and analyses beyond the results presented in the main text (Sec. 4) are discussed here.

## D.1  QUANTITATIVE MODEL EVALUATION

To supplement the qualitative evaluation of visually assessing representative generated samples, a quantitative evaluation based on numerical metrics will provide a more objective approach to compare generation distributions with the real image distribution. Since certain desiderata turn out to be notoriously difficult to quantify (Borji, 2019; 2021), quantitative and qualitative evaluations should be seen as complementary. The Fréchet inception distance (FID) (Heusel et al., 2017) is

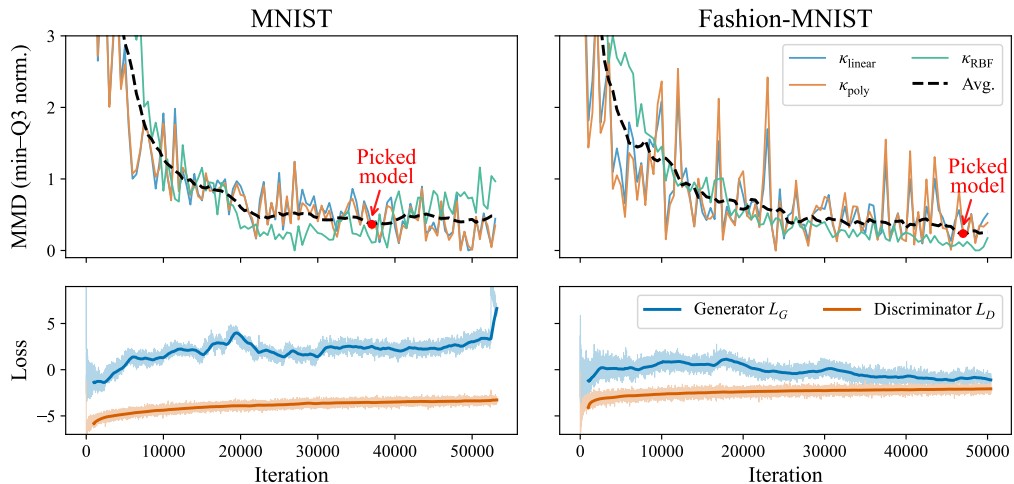

Figure 11: Learning curves of MMD and loss for the largest QGANs (64 layers) on MNIST and Fashion-MNIST. The MMD metric is normalized using its minimum (min) and upper quantile (Q3). The average MMD curve is computed across the three kernels $\kappa_{\text{linear}}$, $\kappa_{\text{poly}}$, and $\kappa_{\text{RBF}}$, followed by a centered moving average over 9 neighboring training checkpoints. The selected model is indicated at the point where the average MMD reaches its minimum. For clarity, the loss curves are additionally smoothed with a moving average over 1 000 iterations.

a widely adopted metric (Borji, 2019; 2021), which was introduced as an improvement over the inception score (IS) (Salimans et al., 2016). Both are briefly summarized below.

**Inception score (IS).** The IS (Salimans et al., 2016) evaluates the quality and diversity of generated images by using the pretrained *Inception-v3 net* (Szegedy et al., 2015) (trained on *ImageNet* (Russakovsky et al., 2015) of 1000 classes and varying resolutions, but well beyond $32 \times 32$ pixels). It measures how confidently the network predicts class labels for each generated image $x$ and how diverse these predictions are across samples. This is based on the *Inception-v3* output node probabilities, where the confidence is reflected through the information-theoretic entropy and relative entropy (KL divergence) measures. A higher IS is better.

**Fréchet inception distance (FID).** Unlike IS, FID compares real and generated images directly in the Inception-v3 feature space (i.e., the activations of a hidden layer, layer $2\,048$ by default), by modeling the activation distributions over the real ($r$) and generated ($g$) image distributions as Gaussians and computing the Fréchet (Wasserstein-2) distance between their means and covariances. For activation means $\boldsymbol{\mu}_r, \boldsymbol{\mu}_g$ and covariances $\Sigma_r, \Sigma_g$,

$$\text{FID} = \|\boldsymbol{\mu}_r - \boldsymbol{\mu}_g\|^2 + \text{Tr}\Big(\Sigma_r + \Sigma_g - 2(\Sigma_r \Sigma_g)^{1/2}\Big). \tag{17}$$

A lower FID is better.

Certain limitations of FID are known and must be considered: first, as the FID is defined via distance measures with respect to the real reference dataset (distribution), FID scores on different datasets are not necessarily comparable. This argument even extends to different subsets of the same dataset. Second, FID has been shown to be sensitive to imperceptible changes, e.g., differences in artifacts arising from image compression (Parmar et al., 2022). Finally, by involving the Inception net trained on ImageNet, a dataset of color photos typically of much larger resolution than the images studied here, FID is typically applied to assess natural color image generation and is potentially less well suited for MNIST and Fashion-MNIST.

## D.2 MODEL SELECTION

All samples presented in this work are generated by either a QGAN after being trained for a fixed number of iterations, or a QGAN reloaded from a training checkpoint, which is selected automatically via the maximum mean discrepancy (MMD) metric instead of the lowest-loss checkpoint, a common criterion in generative modeling (Borji, 2019; 2021). Generally, the number of iterations is set before training starts, or training is stopped after a preset time limit, independent of the loss or evaluation metrics. Importantly, in both model selection scenarios, human intervention or selection was not involved to avoid biased or "cherry-picked" results.

**Maximum mean discrepancy (MMD).** The kernel MMD (Gretton et al., 2012) measures the difference between two probability distributions $\mathbb{P}_{\boldsymbol{x}}$ and $\mathbb{P}_G$, denoting the real data distribution and generator distribution, respectively, in the context of QGAN evaluation. Intuitively, MMD compares similarities within and across datasets, providing a measure of how well the generator mimics the real distribution. For the largest QGAN models in this work, which were used to generate the images in Figs. 3a and 3b, the learning curves of MMD and loss are presented in Fig. 11. The empirical definition of the MMD, based on $k$ samples $\boldsymbol{x}^{(1)}, \dots, \boldsymbol{x}^{(k)} \sim \mathbb{P}_{\boldsymbol{x}}$ (a random $k$-sized subset of the training set) and $\hat{\boldsymbol{x}}^{(1)}, \dots, \hat{\boldsymbol{x}}^{(k)} \sim \mathbb{P}_G$, reads as follows

$$\mathrm{MMD}_\kappa = \frac{1}{k^2} \sum_{i,j} \kappa(\boldsymbol{x}_i, \boldsymbol{x}_j) - 2\frac{1}{k^2} \sum_{i,j} \kappa(\boldsymbol{x}_i, \hat{\boldsymbol{x}}_j) + \frac{1}{k^2} \sum_{i,j} \kappa(\hat{\boldsymbol{x}}_i, \hat{\boldsymbol{x}}_j), \tag{18}$$

where $\kappa$ denotes the kernel. The kernel $\kappa$ is a symmetric similarity function assigning high values to similar samples and low values to dissimilar ones. We evaluate MMD using three common kernels: *linear*, *polynomial* (of degree 2), and *radial basis function (RBF)* (with unit bandwidth) kernels. To obtain stable scores, the MMD values are normalized between their minimum and upper quantile for each kernel, avoiding sensitivity to noisy estimates from early underfit stages. The final score to pick the best model is computed by averaging across kernels and applying a centered moving average (window size 9) across neighboring training checkpoints. A checkpoint was created every 500 iterations, and $k = 5000$ samples were used to estimate the MMD.

## D.3 IMPACT OF GENERATOR DEPTH AND DATASET COMPLEXITY ON IMAGE QUALITY

An extended analysis is presented here to investigate further the relationship between model depth, dataset complexity (in terms of the number of classes), and image quality. The experiments are based on MNIST, comparing models trained on either the complete set of classes or a restricted subset (digits *0* and *1*), while varying generator depths $L \in 8, 16, 32$. The number of modes is set to either 2 or 10, matching the number of classes. Each model is trained for 40 000 iterations, and the checkpoint minimizing the MMD metric is used for image generation. Figure 12 presents the results and clearly shows that as the number of classes increases, deeper generator circuits are required to maintain image quality. Recall that the large models in Sec. 4.1 used 64 layers to capture all MNIST digits at both high quality and diversity.

For the training restricted to two classes, even a shallow generator with $L = 8$ produces high-quality samples (Fig. 12a). This setting exactly matches the number of layers per patch generator used in prior work by Tsang et al. (2023), where 28 such generators were combined for two classes. In contrast to their results, the model here produces images of improved quality, comparable to the results of the much larger 64-layer generators (cf. Fig. 3a). This demonstrates that the gain in quality over previous works is primarily due to our task-specific QGAN design, not merely due to increasing model size. In comparison, Fig. 12b shows images generated from a model with the same number of layers $L = 8$ but now trained on all 10 classes. Here, a decrease in quality is evident, especially when comparing to the *0* and *1* samples in Fig. 12a. Generally, most other classes are captured considerably worse than by the large model as in Fig. 3a. By scaling the model to $L = 16$ (Fig. 12c) and $L = 32$ (Fig. 12d) layers, a successive increase in image quality can be observed. While some images at $L = 16$ (Fig. 12c) already reach a high quality, such as digits *0* and *1* again matching the high quality of the smaller $L = 8$ model when trained on these two classes only (Fig. 12a), it requires $L = 32$ layers (Fig. 12d) to achieve uniformly such quality across all classes.

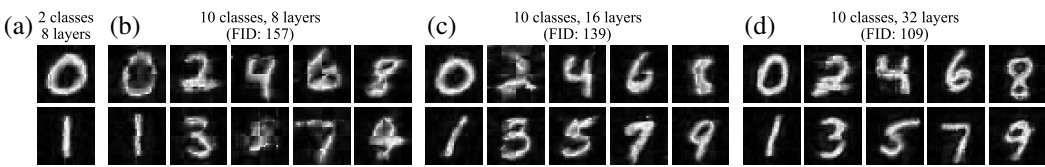

(a) 2 classes 8 layers  (b) 10 classes, 8 layers (FID: 157)  (c) 10 classes, 16 layers (FID: 139)  (d) 10 classes, 32 layers (FID: 109)

Figure 12: Effect of generator depth and dataset complexity on image quality. Models were trained either on (a) an MNIST subset (digits *0* and *1*) with depth $L = 8$, or on all ten MNIST classes with (b) the same depth $L = 8$ or increased depths of (c) $L = 16$ and (d) $L = 32$. One representative image (manually selected) per class is depicted. Results suggest that increasing the number of classes requires deeper generators to maintain visual quality. FID omitted for (a) due to limited comparability between data subsets.

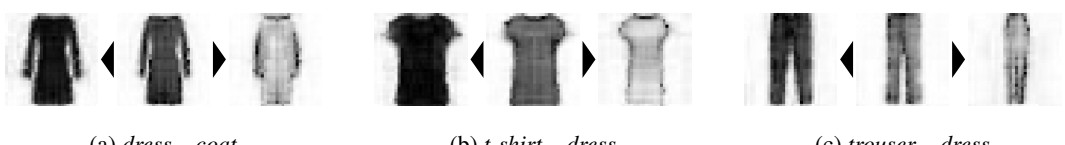

(a) *dress – coat*  (b) *t-shirt – dress*  (c) *trouser – dress*

Figure 13: Three inter-class modes blend between Fashion-MNIST classes: (a) *dress–coat*, (b) *t-shirt–dress*, and (c) *trouser–dress*. Results are from training a QGAN with 4 times more input noise modes than dataset classes ("overmoding"). As in Fig. 6, each mode is visualized with images representing the mean (center) and $\pm 3\sigma$ variations (outer) along the first principal component. PCA is based on 1 000 samples per mode. Note that (a) *dress–coat* mode was already presented in Fig. 6c.

### D.4 INTER-CLASS MODES IN OVERMODING

In extension of the analysis on QGAN "overmoding" for the complete Fashion-MNIST dataset, Fig. 13 presents additional modes, analogous to those in Fig. 6. Here, the focus is on *inter-class* modes, which capture images blending between two classes, occurring in the model trained with 40 input noise modes on the 10 classes in Fashion-MNIST. While such blending may initially appear to induce undesired mixing artifacts, it can in fact reflect realistic scenarios. For instance, one mode morphs between *dress* and *coat*, not only adjusting the shape but also introducing a clear line for a zipper (Fig. 13a). Another mode mostly captures *t-shirts* gradually transitioning from a fitted t-shirt into a t-shirt *dress* (Fig. 13b). A third mode gradually brings the legs of a *trouser* closer together until they eventually connect and resemble a *dress*, while the top simultaneously forms proper shoulder caps (Fig. 13c).

### D.5 LAYER-WISE GENERATOR ANALYSIS VIA SUBSYSTEM ENTROPIES

To gain additional insight into how the generator circuits progressively construct an FRQI image state, we analyze the development of entanglement across different qubit subsets during training. Specifically, we compute subsystem (von Neumann) entropies layer-by-layer for selected groups of qubits in an 11-qubit generator (one color qubit and ten address qubits) trained on the binary MNIST subset. For a quantum state $\rho$, with $\rho = |\psi\rangle\langle\psi|$ if pure, the reduced state on an index set of qubits $\mathcal{A}$ is denoted $\rho_{\mathcal{A}} = \mathrm{Tr}_{\bar{\mathcal{A}}}(\rho)$, where $\mathrm{Tr}_{\bar{\mathcal{A}}}$ is the partial trace over $\bar{\mathcal{A}}$ as the complement of $\mathcal{A}$. The subsystem entropies are the usual von Neumann entropy

$$S(\rho_{\mathcal{A}}) = -\mathrm{Tr}[\rho_{\mathcal{A}} \log_2 \rho_{\mathcal{A}}], \tag{19}$$

which quantifies the entanglement between qubits $\mathcal{A}$ and the rest. Increasing entropy indicates higher levels of entanglement, with a maximum of $|\mathcal{A}|$ bits and a minimum of 0 bits, which certifies no entanglement.

Figure 14 tracks three such entropies over the 8 layers (2 sub-layers) of a 11-qubit generator trained in our QGAN framework on binary MNIST: color qubit $\rho_{\{1\}}$, odd address qubits $\rho_{\{1,2,4,6,8,10\}}$, which address the same spatial dimension via FRQI Morton order, and the two finest resolution address qubits $\rho_{\{10,11\}}$. Note that we include the color qubit in the second subset, i.e., odd address

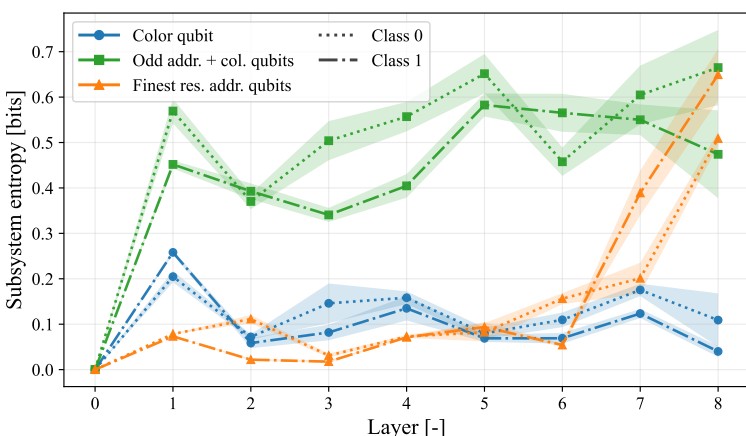

Figure 14: Layer-wise subsystem entropies for different qubit subsets in a generator trained on the binary MNIST dataset. The entropies of the color qubit, odd address qubits (addressing same spatial dimension via Morton order), and two finest resolution address qubits are tracked over the 8 layers of the generator. Layer 0 reflects the initial (unentangled) product state. Note that the color qubit is included with the odd address qubit subsystem. The mean entropy with bands representing the standard deviation across 500 sample generations are depicted. The evolution of these entanglement structures mirrors the emergence of different image features.

and color qubits. We interpret the entropy trajectories and identify key relations to the progressive development of image properties throughout the generation process:

- **Early entanglement for color and spatial control.** The color and odd address show early entanglement, especially in a sharp increase after the first layer. Hence, the generator gains control over the pixel color at specific positions (since color qubit entropy increases). More specifically, the generator correlates between the two spatial dimensions, because all odd address qubits, which address a single spatial dimension, become entangled with address qubits of the other spatial dimension. This is reflected in the increased entropy of the odd address (and color) qubits. For instance, when coloring a single image quadrant, where the color depends on both the x- and y-pixel positions, differently from the rest, the corresponding address qubits (at the first resolution level) must become entangled with other qubits, e.g., $|\psi\rangle = (|100\rangle + |001\rangle + |010\rangle + |011\rangle)/2$ for a white top-left quadrant in an otherwise black image.

- **Coarse-to-fine generation process.** The fine-resolution address qubits exhibit low entanglement throughout most layers, with a sharp increase in entropy towards the final layers. This indicates that visual details are gradually introduced as the generator progresses, similar to a painting process where background and coarse shapes are laid out first, followed by the addition of fine details.

- **Progressive impact of noise.** Noise effects become more pronounced later in the training process, as indicated by higher standard deviations in entropy across different sample generations. In contrast, earlier layers exhibit more deterministic behavior, with nearly zero standard deviation initially. This observation holds because the generator employs bimodal noise inputs (including learnable noise tuning), and we split the visualization by mode, which faithfully represents the sharp bimodal distribution of the entropies. As a result, within each mode, the generator establishes a consistent foundational entanglement structure that captures the basic geometric shapes of each digit, before the sample individualizes through the injected noise.

- **Class-specific differences.** For digit *0* samples, entropy fluctuations (standard deviations) are higher, likely reflecting the greater diversity of shapes within this class. In contrast, digit *1* samples exhibit lower mean entropy values in the odd address qubit subsystem, implying that less entanglement is required to capture color correlations along the image $y$-axis. This could be because *1*s are typically more vertically ($y$-axis) aligned, requiring fewer spatial correlations.

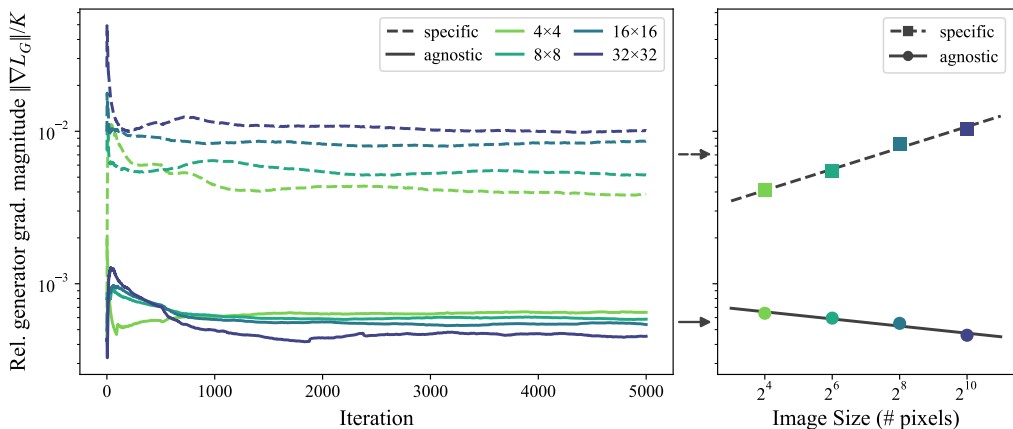

Figure 15: Scaling of the relative generator gradient magnitude $\|\nabla L_G\|/K$ (i.e., normalized by the number of parameters $K$) across MNIST images (digits *0* and *1*) of different resolutions: $4 \times 4$, $8 \times 8$, $16 \times 16$, $32 \times 32$ (light green to dark blue). Two generator designs are shown: image-specific (dashed), with 8 layers (2 sub-layers), and agnostic (solid), with 16 layers. Magnitudes are displayed on a log scale. Left panel: relative magnitudes are depicted over training iterations (linear scale). For visual clarity, a moving average with a window size of 500 iterations is shown. Right panel: Mean relative magnitudes, computed over iterations $1\,000$–$5\,000$, plotted against image size in total number of pixels ($\log_2$ scale, making it proportional to the number of qubits used in the quantum image encodings, i.e., FRQI for the image-specific design and amplitude encoding for the agnostic design). Linear trend lines are added for each generator design. It is evident that the relative gradient magnitudes are not only higher on average in the image-specific design than in the agnostic one, but also *increase* with image size in the specific case, whereas they decrease in the agnostic design, suggesting that the image-specific generator is less susceptible to trainability issues associated with the barren plateau phenomenon.

### D.6 Gradient scaling with image size: examining trainability and barren plateaus

To investigate potential trainability issues associated with the barren plateaus phenomenon, we analyze the relative generator gradient magnitude $\|\nabla L_G\|/K$ as a function of image sizes $N$, as it directly ties to the number of qubits $n$ via $N = 2^n$ (plus one qubit for gray-value FRQI). Figure 15 shows these magnitudes for the proposed image-specific generator design vs an agnostic one on an MNIST subset (digits *0* and *1*). The gradient signal never vanishes throughout training and stabilizes after around $1\,000$ training iterations, while the magnitudes are generally higher for the specific than for the agnostic generators. The agnostic generator exhibits decreasing magnitudes, indicating a higher susceptibility to vanishing gradients (barren plateaus) as the problem (image) size grows. This is consistent with expectations for generic ansätze lacking inductive bias (Larocca et al., 2025).

In contrast, the image-specific generator maintains larger gradients that even *increase* with image size. Importantly, this observed increase in gradient magnitude per parameter in the specific design may be unexpected considering an increasing problem size and, hence, number of qubits. This observation must be interpreted in light of the constant number of layers. It suggests that the number of layers could potentially be scaled proportionally with the number of qubits (as typically required to achieve finer image details at higher resolutions) while maintaining the gradient signal or limiting its decay. Preliminary experiments with a non-constant depth, scaled proportionally to the image size, indicate that the mean gradient magnitude per parameter no longer increases with problem size but instead quickly approaches a constant value, providing preliminary validation of our trainability claim.

While this analysis provides some preliminary empirical evidence, more rigorous investigations of potential barren plateaus would require a theoretical treatment. Recent work on barren plateaus in generative quantum machine learning, similar to, e.g., Rudolph et al. (2024); Letcher et al. (2024),

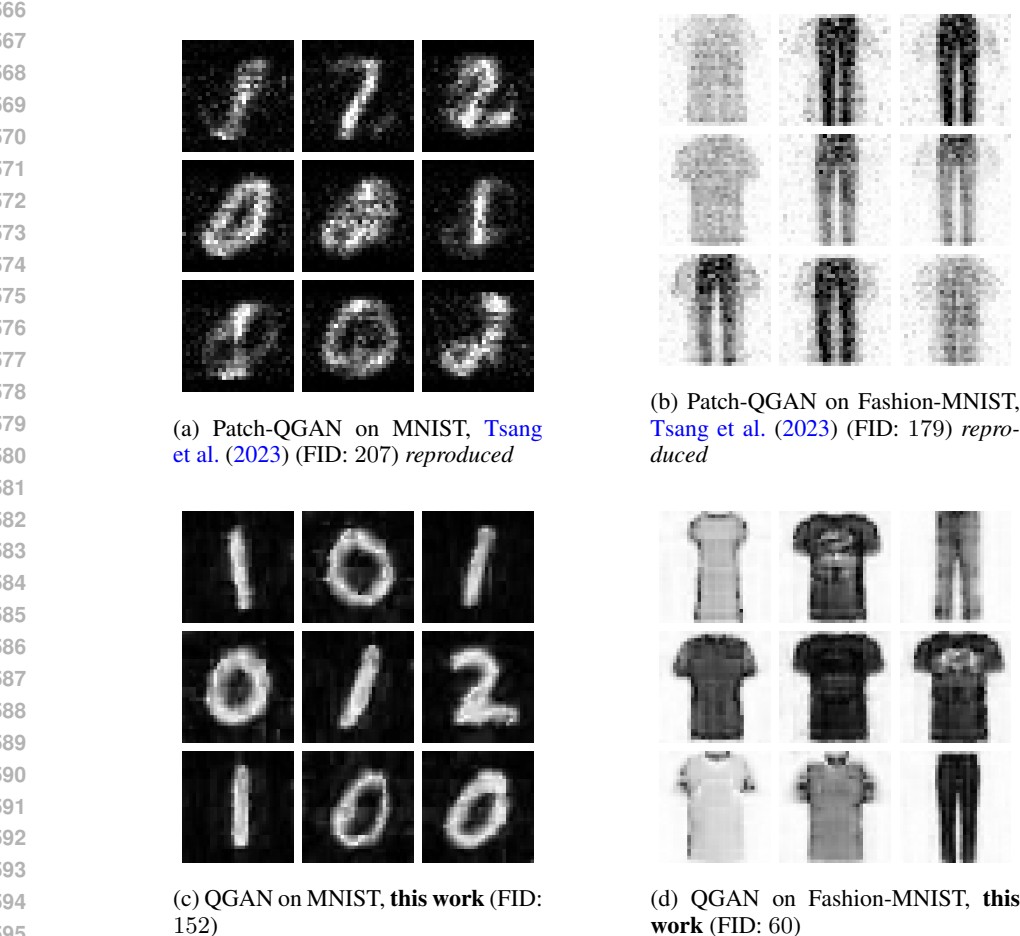

(a) Patch-QGAN on MNIST, Tsang et al. (2023) (FID: 207) *reproduced*

(b) Patch-QGAN on Fashion-MNIST, Tsang et al. (2023) (FID: 179) *reproduced*

(c) QGAN on MNIST, **this work** (FID: 152)

(d) QGAN on Fashion-MNIST, **this work** (FID: 60)

Figure 16: Random samples from the (a-b) patch-QGAN of Tsang et al. (2023) and (c-d) our full-image QGAN framework. Subsets of (a, c) MNIST (*0/1/2*) and (b, d) Fashion-MNIST (*t-shirt* and *trousers*) are considered. Our QGAN approach yields visually cleaner images, as also reflected in the decrease in FIDs reported.

underscores the need for formal analysis beyond empirical trends. Although the present study offers promising early indications, it remains constrained by the use of image data of only moderate size.

## D.7 BENCHMARKING WITH PATCH-GENERATION QGANS

As discussed in the main text, we identify the patch-generation QGAN framework (Tsang et al., 2023) as the state-of-the-art baseline in quantum generative learning for which a quantum model without heavy classical postprocessing is used to generate the images. We compare our QGAN approach, where one generator generates the full images, as opposed to patch-QGANs, where one generator is employed per image row. We adopt the largest subsets of MNIST and Fashion-MNIST, namely three digits and two classes, respectively, as covered by Tsang et al. (2023), to train patch-QGANs in the exact same, most expressive designs and configurations: 28 row-patches, 7 qubits, 11 layers, which results in a total number of 7 392 parameters in the generation. Similarly, we train the patch-QGANs for about 25 and 37.5 epochs on MNIST and Fashion-MNIST, respectively, and subsequently sample 10 000 images to compute the FIDs. Random subsets of these samples are presented in Figs. 16a (MNIST) and 16b (Fashion-MNIST). We fully reproduce the models based on the corresponding implementation on GitHub[1] and the MNIST and Fashion-MNIST datasets in the original $28 \times 28$ resolution. Refer to Tsang et al. (2023) for further details.

---

[1] https://github.com/jasontslxd/PQWGAN

For MNIST, a direct visual comparison between patch-QGAN (Fig. 16a) and our QGAN framework (Fig. 16c) clearly shows the superior generation capabilities, reflected in both clear, noise-free image quality without class blending artifacts in our results. Note that the 16-layer QGAN model trained for the ablation study (presented initially in Fig. 4d in Sec. 4.2) provides the images for the comparison here. Quantitatively, the corresponding FIDs significantly favor the results produced by our QGAN (FID 152) over the path-based one (FID 207).

For Fashion-MNIST, the observations from MNIST directly transfer, both qualitatively, comparing Figs. 16b (patch-QGAN) and 16d (our QGAN framework), and quantitatively with a significant decrease in FID of 179 for the patch-QGAN results to 60 in our QGAN approach. Note that we did not retrain a QGAN model in our framework on the 2-class subset of Fashion-MNIST. Instead, we present only samples (and consider them for the FID reported here) from the modes corresponding to the two classes *t-shirt* and *trousers*, which is clear through visual inspection. This treatment should not give our QGAN any advantage. On the contrary, it may cause mixing with characteristics of other classes into different modes (as discussed in the inter-class mode study in Section D.4). This could result in out-of-distribution samples with respect to the two-class FashionMNIST subset. Given the significant differences in FIDs, a minor bias should not affect the reliability of this comparison.

## LLM USAGE STATEMENT

In accordance with the ICLR 2026 policy on Large Language Model (LLM) usage, we disclose that Grammarly and ChatGPT were utilized for grammar checking, style improvement, and minor text polishing. GitHub Copilot was used to suggest code snippets during development, with all generated code reviewed, tested, and adapted by the authors. No LLMs were used for generating novel research ideas, data analysis, or drafting substantial portions of the manuscript.

