# OpenReview forum: "Scaling Quantum Machine Learning without Tricks: High-Resolution and Diverse Image Generation"
_ICLR.cc/2026/Conference — Submitted to ICLR 2026_

### Official Review · Reviewer_5cw3 · 2025-10-23

**Soundness:** 3
**Presentation:** 2
**Contribution:** 2
**Rating:** 4
**Confidence:** 3

**Summary:**

This paper presents QGAN, a single, end-to-end quantum circuit based generative framework capable of generating high-quality, diverse images. It moves beyond previous approaches that relied on classical dimensionality reduction or patch-based generation, which limited the quantum model's role and overall performance.

**Strengths:**

1) This work introduces that a single quantum generator can generate diverse images without relying on patches or dimensionality reduction.

2) It proposes a structured ansatz for image generation, moving beyond generic, task-agnostic circuits.

3) It introduces a learnable, multi-modal latent space for the generator leveraging reparametrization tricks, enabling generation of diverse, high-quality images.

3) Extensive experiments are performed mimicing realistic conditions, e.s.p. the rebustness study.

**Weaknesses:**

1. Clarity of the work needs to be improved, e.s.p. regarding the model architecture. The current explanation of the model make it hard to intepret and understand. A more direct visualization of the model is helpful.

2. Lack of theoretical analysis (or advantage)  and comparisons with the existing methods. Without theoretical analysis, it is hard to understand what quantum properties can help the generative process.

**Questions:**

See weakness.

**Details Of Ethics Concerns:**

No.

---

> ### Author Response · Authors · 2025-11-20
> **Official Response to Reviewer 5cw3 (part 1)**
>
> We express our gratitude to the referee for their time and helpful feedback on our work. We are glad the referee recognized several convincing strengths in our work. In light of the identified weaknesses and questions raised by all referees, we provide a revised version of our work. Before directly addressing the referee's specific comments, we present a list of major modifications:
>
> - Ground-up revision of Method (Section 3) plus expanded modular schematic (Fig. 1).
> - Added quantitative evaluation via Fréchet Inception Distance (FID) metric for relevant experiments
> - Added benchmarks/comparisons: both quantum (patch-generation QGAN (Tsang 2023)) and classical (WGAN-GP), see Discussion (Section 5)
> - Added inductive bias rationale theoretically (Appendix B.2.2) and numerically (Appendix D.5)
> - Added new analysis on gradient scaling with image size, comparing behavior for image-specific vs agnostic generator design (Appendix D.6).
> - Added model requirements (incl. parameter counts) and quantum resource estimates (compiled circuit depths and CNOT counts) in Appendix B.5, Table 1 for the different generator configurations used.
>
> We continue to address each point regarding weaknesses and questions the referee explicitly raised. We first recite the original statement, then provide the reply.
>
> ## Weaknesses:
>
> > 1. Clarity of the work needs to be improved, e.s.p. regarding the model architecture. The current explanation of the model make it hard to intepret and understand. A more direct visualization of the model is helpful.
>
> We thank the referee for highlighting the potential for improvement in the clarity of the work. We are glad to address this by improving the presentation and structure through a ground-up revision of the Method section (Sec. 3), including an extension of the schematic in Fig. 1 to include a modular overview of the complete QGAN framework, highlighting the generator design and its integration into the complete QGAN training workflow (including the discriminator and data) more clearly. Moreover, we would like to emphasize that the codebase implementing the full QGAN framework is provided (see reproducibility statement) to retain details on the model architecture.

---

> ### Author Response · Authors · 2025-11-20
> **Official Response to Reviewer 5cw3 (part 2)**
>
> ### Weaknesses (cont'd):
>
> > 2. Lack of theoretical analysis (or advantage) and comparisons with the existing methods. Without theoretical analysis, it is hard to understand what quantum properties can help the generative process.
>
> We thank the referee for raising this vital point regarding theoretical analysis and comparisons with existing methods. We would like to emphasize that we do not claim a quantum advantage in this work, nor aim to outperform classical approaches (we added a clarifying statement on this to the Discussion in lines 482f and 504f). Our focus is on systematic empirical evaluation of our QGAN framework. Specifically, we provide ablation studies investigating generator design choices, showing that the proposed ansatz exhibits favorable scaling properties compared to agnostic designs.
> To understand how quantum properties such as entanglement are reflected in the generation process, we added a full study in Appendix D.5 that analyzes the development of entropy in different subsystems throughout the generation process, i.e., layer by layer, and relates them to image properties.
>
> To directly address the referee’s concern regarding comparisons with existing methods, we added comprehensive comparisons. The revised manuscript now includes comparisons with prior quantum and classical GANs:
>
> 1. Quantum GANs: As already motivated earlier, the patch-generation QGAN (Tsang et al., 2023) provides an appropriate baseline for our QGAN approach. We methodologically improve upon their approach and tackle the more challenging problem of directly generating full images instead of low-dimensional patches. Quantitative comparisons via Frechet Inception Distances (FIDs) are discussed in the Discussion (Section 5), while visual (qualitative) comparisons are provided in Appendix D.7. Note that other prior QGAN methods often feature highly hybrid generator implementations (i.e., significant classical postprocessing, see "Trick 1" in Introduction), where the contribution of the generator beyond a better noise source is often unclear. Hence, did not consider such approaches for a direct comparison.
> 2. Classical GANs: We relate our QGAN results quantitatively (via FID) to the classical analogous gradient-penalized Wasserstein-GAN. Here, we rely on a large-scale benchmark by Lucic et al. (2018) and their reported metrics for MNIST and Fashion-MNIST. All details are provided in the Discussion (Section 5).
>
> Regarding theoretical analysis, a rigorous formalization of image properties and their relation to quantum states is currently lacking in both classical and quantum generative modeling. While we agree that such analyses would be desirable, they are often not obvious in either classical ML or QML. As in classical ML, empirical study is a crucial tool, and theoretical proofs for generative models are rare. For example, we nevertheless briefly added a discussion of how different families of FRQI image transformations can be learned in our generator ansatz in Appendix B.2.2, offering preliminary theoretical motivation for the generator ansatz and its alignment (i.e., inductive bias) with FRQI.
>
> ## Questions:
>
> No additional questions were raised.

---

> > ### Comment · Reviewer_5cw3 · 2025-11-26
> >
> > I thank the reviewers for the reply. Considering other reviewer's opinion, I would maintain my current score.

---

> > > ### Author Response · Authors · 2025-11-27
> > > **Authors’ Follow-Up to Reviewer 5cw3**
> > >
> > > We thank the referee again for the careful initial review and the update acknowledging our rebuttal. We truly appreciate the time invested and are grateful for the constructive feedback that helped us improve the manuscript substantially.
> > >
> > > The referee's update states that they would like to maintain the original score of 4, considering other reviewers' opinions. We respectfully believe that the major revisions uploaded on Nov 20 substantially resolve the points raised by the referee. To make this easy to verify, here is a concise mapping:
> > >
> > > - **Clarity & visualization of method**: Section 3 completely rewritten from the ground up; Fig. 1 extended into a full modular overview of the QGAN workflow (initial noise ↔ generator ↔ discriminator & data).
> > > - **Comparison & theoretical properties**: Explicit comparison with both quantum and classical approaches (Discussion, lines 482-506, and Appendix D.7), demonstrating substantial quantitative and qualitative improvements over the previous state-of-the-art quantum baseline. While our work is of systematic empirical nature, we add concise theoretical support for our ansatz design relating it to FRQI image operations (Appendix B.2.2), supplemented by entanglement analysis in generation process (Appendix D.5, Fig. 14).
> > >
> > > These changes directly respond to every weakness and question the referee listed, and we believe they transform the earlier concerns about clarity, reproducibility, novelty, and relevance.
> > >
> > > We would be very grateful if the referee could take a quick look at the revised version again and let us know whether, in light of these concrete additions, the remaining concerns are resolved and a score adjustment feels warranted. Otherwise, we would invite the referee to share open concerns or questions that justify the current score, and we are willing to provide a new revision of our manuscript.
> > >
> > > We thank the referee once again for their expertise and for helping us strengthen the paper. We are happy to add any further details and clarifications.

---

> > > > ### Comment · Reviewer_5cw3 · 2025-11-28
> > > >
> > > > My overall impression is, as pointed out by the Reviewer o4r5, is: 1) the measurement cost in the generation process. Per the definition of quantum generative models, it should be that every shot gives a sample or least a descriptor which can be used for generation, instead of expectation values. The expectation value from of the quantum circuit measurement has clear definitions and mathematical expressions in Fourier domain for encoded data. Therefore, I dont see how this translates to generative models and give an advantage; I would not doubt that it can do this generative task though. Secondly, the generated image quality that has been shown in this paper, in my opinion, is still quite low. Therefore, I am still not sure if this is the correct path for generative model. I would hope that the author could provide more convincing evidence in such designs and improve the manuscript following the previous advice.

---

### Official Review · Reviewer_LScu · 2025-11-01

**Soundness:** 3
**Presentation:** 3
**Contribution:** 3
**Rating:** 6
**Confidence:** 4

**Summary:**

Previous quantum machine learning works (especially quantum generative models, as considered in this work) are often with limited size due to difficulty of data uploading and training. As an example, for the MNIST datasets, many previous works first reduce the dimension of images. By introducing inductive bias created by an application-specific quantum circuit design, this work is able to train a quantum GAN without reducing any dimension.

**Strengths:**

Scaling is one of the most important issue regarding to the quantum machine learning models. Though heuristic, it is good to see that the work makes some positive progress towards this point. The overmoding, i.e., more input noise modes, inspired by the classical over-parameterization, seems to be interesting. Shown as Fig.6 in the paper, the numerical result agrees with the authors' argument that increasing mode can increase the model performance. The size for QGAN models with 64 layers and 40 noise modes for about 50 000 generator updates is also relatively big compared with other QNN models.

**Weaknesses:**

The weakness of this paper is that it does have any theoretical analysis to support their claim, and also the method is model-specific, and thus it is not that clear how good this method will be.

**Questions:**

1. I would like the authors to clarify more about the model-specific trick. What properties or structure are essentially required for the input data set?
2. For the overmoding, is it possible to provide more theoretical aspect and seek any potential for a quantum advantage?
3. Hope to see a bit more discussion about the scalability based on the numerical result.

---

> ### Author Response · Authors · 2025-11-20
> **Official Response to Reviewer LScu (part 1)**
>
> We express our gratitude to the referee for their time and helpful feedback on our work. We are glad the referee recognized several convincing strengths in our work. In light of the identified weaknesses and questions raised by all referees, we provide a revised version of our work. Before directly addressing the referee's specific comments, we present a list of major modifications:
>
> - Ground-up revision of Method (Section 3) plus expanded modular schematic (Fig. 1).
> - Added quantitative evaluation via Fréchet Inception Distance (FID) metric for relevant experiments
> - Added benchmarks/comparisons: both quantum (patch-generation QGAN (Tsang 2023)) and classical (WGAN-GP), see Discussion (Section 5)
> - Added inductive bias rationale theoretically (Appendix B.2.2) and numerically (Appendix D.5)
> - Added new analysis on gradient scaling with image size, comparing behavior for image-specific vs agnostic generator design (Appendix D.6).
> - Added model requirements (incl. parameter counts) and quantum resource estimates (compiled circuit depths and CNOT counts) in Appendix B.5, Table 1 for the different generator configurations used.
>
> We continue to address each point regarding weaknesses and questions the referee explicitly raised. We first recite the original statement, then provide the reply.
>
> ## Weaknesses:
>
> > The weakness of this paper is that it does have any theoretical analysis to support their claim, and also the method is model-specific, and thus it is not that clear how good this method will be.
>
> We thank the referee for raising this two-fold point. First, we agree with the referee that theoretical analyses are generally important and desirable, but they are often not obvious, which is what makes ML an empirical field, and so QML. There are very few theoretical proofs for classical generative models.  Hence, our focus is on a systematic empirical evaluation of our QGAN framework, which is further extended by additional experiments and analyses (Appendix D) in the revised manuscript. Also, we provide a direct comparison with a previous state-of-the-art QGAN baseline (i.e., patch-generation QGANs; Tsang et al., 2023), quantitatively evaluated in the Discussion (Section 5), and qualitative details provided in Appendix D.7. Our QGAN not only tackles the more challenging problem of directly generating full images instead of low-dimensional patches, but outperforms patch-generation QGANs clearly.
>
> Second, for the concern regarding the specific design, having a specific generator design over a problem-agnostic one is exactly the key feature of the approach, not a weakness. The design encodes an inductive bias towards image data in general and, importantly, not the specific image datasets like (Fashion-)MNIST, and could hence be readily applied to other image datasets that were not covered in our work without compromising the performance expected.
> Moreover, this approach mirrors lessons from classical deep learning, where models are also typically less agnostic/generic than they might seem. For instance, convolutional neural networks (CNNs) exhibit specialized architecture tailored towards image processing (even inspired by the visual processing in brains) rather than a generic (theoretically universal) feed-forward neural network. Similarly, our generator architecture demonstrates that careful, problem-specific design is crucial for advancing practical QGAN image generation.

---

> ### Author Response · Authors · 2025-11-20
> **Official Response to Reviewer LScu (part 2)**
>
> ## Questions:
>
> > 1. I would like the authors to clarify more about the model-specific trick. What properties or structure are essentially required for the input data set?
>
> We thank the referee for raising this question. We identify two main features of image input data sets that inform the design choices for our quantum generator. First, the multimodal nature of distributions over image data. We exemplified this with a center pixel in MNIST subset for digits 0 and 1 in the histogram provided in Figure 2 (right side). We directly facilitate multimodal learning by providing a multimodal latent distribution as the generator's noise input. We empirically verify the importance of multimodality (and its injection, i.e., via noise tuning) in the experiments (Section 4.2, esp. Figure 5).
> Second, natural images exhibit strong spatial correlations: neighboring pixels tend not to vary independently but form coherent regions, edges, and textures. Our ansatz is designed to reflect this property by aligning the circuit structure with the FRQI qubit layout (including the Morton/Z-order address mapping).
> Regarding the image-specific generator ansatz, we provide theoretical arguments by relating typical families of FRQI image transformations to fit our ansatz design (Appendix B.2.2), i.e., encode inductive bias into the ansatz, and added a study (Appendix D.5) on how quantum properties such as entanglement develop throughout the generation process and how they correspond to image properties.
>
> Moreover, data needs to be compressible, as is usually the case for useful classical data, e.g., images, time series, etc. Compressibility enables efficient loading into a quantum computer and, conversely, efficient generation, i.e., moderately shallow circuits under such amplitude-type encodings like FRQI.
>
> > 2. For the overmoding, is it possible to provide more theoretical aspect and seek any potential for a quantum advantage?
>
> We thank the referee for this interesting question and highly appreciate the deep thoughts about our work that are implied by this question. In fact, we considered a potential advantage through overmoding by replacing the classical bits for the input mode $m$ with qubits (when interpreting the multimodal noise injection via conditional rotation gates per mode controlled by the classical mode bits as depicted in Figure 2). One could, in principle, model an exponential number of modes with respect to these ancilla qubits (initialized in equal superposition) if they are non-trivially entangled with the remaining FRQI qubits. However, the repeated measurements to obtain a single image sample must then be restricted to one mode. While it works under a post-selection assumption, this constrained measurement is otherwise more challenging. Amplitude amplification techniques might help.

---

> ### Author Response · Authors · 2025-11-20
> **Official Response to Reviewer LScu (part 3)**
>
> ### Questions (cont'd):
>
> > 3. Hope to see a bit more discussion about the scalability based on the numerical result.
>
> We thank the referee for this query. We interpret "scalability" here as the ability of the proposed QGAN architecture to extend beyond toy settings and toward higher-dimensional, full-resolution image generation.
>
> First, compared to prior QGAN works, which generally operate only in low-dimensional settings via either (i) classical dimensionality reduction or (ii) patch-based generation, our models scale to the direct generation of full-resolution images. The newly added comparison to the patch-generation QGAN baseline (see Discussion (Section 5) and Appendix D.7) further highlights how our architecture performs in regimes where previous quantum approaches struggle, addressing the referee’s request for a scalability discussion grounded in numerical results.
>
> Second, we now contextualize scalability with respect to classical generators. Classical GANs achieving comparable FIDs on average hyperparameter settings typically use more than two orders of magnitude more parameters than our largest (64-layer) QGAN. This comparison, added to the Discussion (Section 5), highlights scalability through the lens of parameter efficiency. For more details on parameter counts in our models, please refer to the newly added Table 1 in the appendix, which also includes quantum resource estimates in terms of compiled circuit depths and CNOT counts.
>
> Third, although the referee notes that our 64-layer model is large relative to prior QML work, we would like to emphasize that strong performance already appears at smaller scales. For MNIST, results from 8-, 16-, and 32-layer models are provided in Appendix D.3, and for Fashion-MNIST, similar trends appear in the overmoding study (Fig. 6c; 32-layer model). These results indicate that scalability to full-resolution images at good quality is not dependent on extremely deep circuits.
>
> Finally, we added experiments under scaling the image size (and hence number of qubits) and studying the gradient magnitude per parameter. This is important for identifying potential trainability issues (e.g., related to barren plateaus) that may arise as problem (i.e., image) size increases. We observe significantly more desirable behavior in our specific generator design than in the agnostic one. See Appendix D.6.

---

> > ### Comment · Reviewer_LScu · 2025-11-26
> > **Reply**
> >
> > Thanks for the reply.
> > The authors partially solve my questions, and I would like to keep my score.

---

> > > ### Author Response · Authors · 2025-11-26
> > > **Thank you**
> > >
> > > Thank you for the follow-up and for considering our clarifications and the revised manuscript. Please feel free to let us know if any additional clarification would be helpful. We are happy to provide any further details and explanations.

---

### Official Review · Reviewer_u5rB · 2025-11-03

**Soundness:** 2
**Presentation:** 2
**Contribution:** 2
**Rating:** 4
**Confidence:** 4

**Summary:**

This paper presents a novel approach to quantum generative modeling by training a single end-to-end quantum Wasserstein GAN (QGAN) on full-resolution datasets such as MNIST, Fashion-MNIST, and SVHN. Unlike prior methods that rely on dimensionality reduction or patch-based generation, the authors leverage application-specific quantum circuit designs and enhanced multimodal noise input to achieve high-quality and diverse image generation. The work demonstrates scalability, improved fidelity, and robustness under quantum shot noise, setting a new benchmark for quantum image synthesis.

**Strengths:**

1. One key strength of this paper is the extensive experimental validation across multiple full-resolution datasets, including MNIST, Fashion-MNIST, and SVHN.

2. The authors demonstrate strong empirical performance, showcasing the scalability and robustness of their quantum generative model.

3. The work also stands out for eliminating classical post-processing and relying solely on quantum-native design, which enhances its novelty and practical relevance.

**Weaknesses:**

The paper lacks a clear and detailed structural presentation of the proposed framework. As a result, the overall methodology remains vague, making it difficult for readers to grasp the model’s design and workflow.

The core approach builds on Quantum Wasserstein GANs, which were introduced by Chakrabarti et al. in 2019 (NeurIPS 32). However, the current work does not offer substantial architectural innovation beyond that baseline, raising concerns about novelty.

Although the authors present extensive experiments, many critical details are missing or ambiguously described. For instance, the computational environment, model configuration, and algorithmic components are not clearly specified. This lack of transparency undermines reproducibility and casts doubt on the robustness of the reported results.

Overall, the paper suffers from weak technical exposition. The narrative is often unclear, and key concepts are buried in verbose descriptions, which makes the model harder to understand rather than illuminating its strengths.

**Questions:**

Could you provide a more detailed schematic or modular breakdown of your proposed QGAN framework? The current description lacks structural clarity, making it difficult to understand how components interact and how the model is trained end-to-end.

Given that Quantum Wasserstein GANs were introduced by Chakrabarti et al. (NeurIPS 2019), what specific architectural or algorithmic innovations distinguish your model from theirs? A clearer comparison would help assess the novelty of your contribution.

Can you elaborate on the experimental setup, including quantum simulator or hardware specifications, number of shots, and noise models used? These details are essential for reproducibility and for evaluating the robustness of your results.

The paper mentions application-specific circuit designs, but does not provide concrete circuit layouts or gate compositions. Could you include representative circuit diagrams or pseudocode to clarify the model’s internal structure?

You mention multimodal noise input as a key design choice. Have you conducted any ablation studies to isolate its impact on generation quality? This would help validate its contribution.

What quantitative metrics were used to assess image quality and diversity? Including standard benchmarks like FID or IS would strengthen your claims and allow comparison with classical GANs.

How feasible is your approach on current noisy intermediate-scale quantum (NISQ) hardware? Please provide resource estimates (e.g., qubit count, circuit depth) and discuss any hardware-specific constraints.

---

> ### Author Response · Authors · 2025-11-20
> **Official Response to Reviewer u5rB (part 1)**
>
> We express our gratitude to the referee for their time and helpful feedback on our work. We are glad the referee recognized several convincing strengths in our work. In light of the identified weaknesses and questions raised by all referees, we provide a revised version of our work. Before directly addressing the referee's specific comments, we present a list of major modifications:
>
> - Ground-up revision of Method (Section 3) plus expanded modular schematic (Fig. 1).
> - Added quantitative evaluation via Fréchet Inception Distance (FID) metric for relevant experiments
> - Added benchmarks/comparisons: both quantum (patch-generation QGAN (Tsang 2023)) and classical (WGAN-GP), see Discussion (Section 5)
> - Added inductive bias rationale theoretically (Appendix B.2.2) and numerically (Appendix D.5)
> - Added new analysis on gradient scaling with image size, comparing behavior for image-specific vs agnostic generator design (Appendix D.6).
> - Added model requirements (incl. parameter counts) and quantum resource estimates (compiled circuit depths and CNOT counts) in Appendix B.5, Table 1 for the different generator configurations used.
>
> We continue to address each point regarding weaknesses and questions the referee explicitly raised. We first recite the original statement, then provide the reply.
>
> ## Weaknesses:
>
> > The paper lacks a clear and detailed structural presentation of the proposed framework. As a result, the overall methodology remains vague, making it difficult for readers to grasp the model’s design and workflow.
>
> We welcome the referee for raising concerns about the clarity of the QGAN framework presentation. We are glad to address this by improving the presentation and structure through a ground-up revision of Sec. 3, particularly by extending the schematic in Fig. 1 to include a modular overview of the complete framework, highlighting the generator design and its integration into the complete QGAN training workflow (including the discriminator and data) more clearly.
>
> > The core approach builds on Quantum Wasserstein GANs, which were introduced by Chakrabarti et al. in 2019 (NeurIPS 32). However, the current work does not offer substantial architectural innovation beyond that baseline, raising concerns about novelty.
>
> We thank the reviewer for raising this important point. We are happy to elaborate on the novelty of our work. While Chakrabarti et al. (2019) indeed introduced the Wasserstein extension in the quantum GAN setting, but in a quantum data application, and Tsang et al. (2023) applied Wasserstein QGANs to classical image generation, this latter work was limited to generating low-resolution image patches. It was explicitly shown to be incapable of scaling to generate full-resolution images directly. Other works had to rely on dimensionality reduction techniques to circumvent the direct generation in high-dimensional image spaces. We extended our work to include a reference to Chakrabarti et al. (2019) (see lines 152-154).
>
> In contrast, our work introduces a problem-specific quantum generator architecture that enables stable training and high-quality, diverse image generation at native resolution across multiple classes. This is a significant step beyond prior approaches, which could not achieve these objectives (e.g., no demonstrations of more than 2 or 3 MNIST classes in the literature, whereas we cover the complete 10-class datasets with a single generator). Our work presents a concrete empirical example of how to address trainability and scalability limitations by replacing agnostic designs with specific ones that induce relevant inductive biases. Thus, although the core theoretical framework builds on WQGANs, the proposed generator architecture and its demonstrated capability on full-resolution, diverse classical image generation represent novel advancements beyond prior work and an essential contribution to quantum GAN research.
>
> Moreover, this approach mirrors lessons from classical deep learning, where the introduction of convolutional neural networks (CNNs) as a more specialized architecture suited for image processing rather than a generic (theoretically universal) feed-forward neural network enabled pivotal advancements in computer vision. Similarly, our generator architecture demonstrates that careful, problem-specific design is crucial for advancing practical QGAN image generation.

---

> ### Author Response · Authors · 2025-11-20
> **Official Response to Reviewer u5rB (part 2)**
>
> ### Weaknesses (cont'd):
>
> > Although the authors present extensive experiments, many critical details are missing or ambiguously described. For instance, the computational environment, model configuration, and algorithmic components are not clearly specified. This lack of transparency undermines reproducibility and casts doubt on the robustness of the reported results.
>
> We thank the referee for emphasizing the importance of reproducibility. We would like to clarify that the full codebase, including exact configuration files for each QGAN experiment, is available and was provided with the original submission. This includes all hyperparameter settings, model architectures, and detailed instructions for fully reproducing our results and analyses. We believe that this complete disclosure not only addresses the referee’s main concern but definitely strengthens reproducibility and undoubly makes our results reliable and robust. Furthermore, we would like to refer to the appendix for the concrete specifications the referee requested, i.e., B (computational environment), B.1 (algorithmic components of GAN training), B.2 (generator design), B.3 (discriminator design), and B.5 (training hyperparameters). Additionally, we included a table (Table 1 in B.5) including all generator model configurations and corresponding resource requirements in the revised manuscript. We kindly welcome any hints to specific settings that appear to be missing, and are more than happy to report them and explicitly include them in the paper/appendix.
>
> > Overall, the paper suffers from weak technical exposition. The narrative is often unclear, and key concepts are buried in verbose descriptions, which makes the model harder to understand rather than illuminating its strengths.
>
> We appreciate the referee for pointing out the weaknesses in our submission, which are mostly related to presentation. We aim to make significant improvements in the revised manuscript to enhance the clarity of the framework and to better emphasize its strengths. We concretely, through a ground-up revision of the Method section (Sec. 3), including an extension of the schematic in Fig. 1 to include a modular overview of the complete QGAN framework, highlighting the generator design and its integration into the complete QGAN training workflow (including the discriminator and data) more clearly. Additionally, the appendix (Appendix D) is expanded to include concrete definitions and specify configurations more clearly.
>
>
> ## Questions:
>
> > Could you provide a more detailed schematic or modular breakdown of your proposed QGAN framework? The current description lacks structural clarity, making it difficult to understand how components interact and how the model is trained end-to-end.
>
> We appreciate the referee for this concrete insight. We are glad to address this by improving the presentation and structure through a ground-up revision of the Method section (Sec. 3), particularly by extending the schematic in Fig. 1 to include a modular overview of the complete framework, highlighting the generator design and its integration into the complete QGAN training workflow (including the discriminator and data) more clearly.
>
> > Given that Quantum Wasserstein GANs were introduced by Chakrabarti et al. (NeurIPS 2019), what specific architectural or algorithmic innovations distinguish your model from theirs? A clearer comparison would help assess the novelty of your contribution.
>
> We thank the referee for raising this question. We address the question in our response to the second raised weakness above.

---

> ### Author Response · Authors · 2025-11-20
> **Official Response to Reviewer u5rB (part 3)**
>
> ### Questions (cont'd):
>
> > Can you elaborate on the experimental setup, including quantum simulator or hardware specifications, number of shots, and noise models used? These details are essential for reproducibility and for evaluating the robustness of your results.
>
> We thank the referee for raising this relevant question for concrete implementation details. Appendix B is very relevant here, where we state that all experiments were conducted via numerical state-vector simulations using the PennyLane Python library with auto-differentiation (for gradient-based optimization via Adam) and GPU-acceleration provided by JAX. Hardware not applicable.
> More details on how shot noise is modelled in our simulations are covered in Appendix B.5.
> No further noise beyond shot noise was considered, i.e., no hardware noise models were included, which is because NISQ applicability is not directly targeted with our QGAN framework due to the relatively deep models, which are beyond current noisy hardware capabilities due to decoherence. Explicit resource estimates regarding compiled circuit depths and CNOT counts are now added to Appendix B.5 in Table 1. Instead, we want to emphasize that we see applicability of our approach for near-term fault-tolerant devices with few logical qubits, as the qubit requirements (11-13 qubits) are moderately low.
> We kindly welcome any hints to specific settings that appear to be missing, and are more than happy to explicitly include them in the paper/appendix.
>
> > The paper mentions application-specific circuit designs, but does not provide concrete circuit layouts or gate compositions. Could you include representative circuit diagrams or pseudocode to clarify the model’s internal structure?
>
> We thank the referee for raising this question. Figure 1 already includes the concrete circuit layouts and gate compositions, including the specific entangling gates used. In addition, all gates are defined in Appendix A, and we have now added an explicit reference to the relevant part of the appendix in the Methods section to make it easier to locate (see line 158).
> To further improve transparency, we also provide circuit layouts and gate compositions for the agnostic circuit baseline in Appendix B.2.1. We would be grateful for any suggestions regarding definitions or details that may still appear unclear, and we would be happy to include them explicitly in the paper or appendix.
>
> > You mention multimodal noise input as a key design choice. Have you conducted any ablation studies to isolate its impact on generation quality? This would help validate its contribution.
>
> We thank the referee for this question. As already included in the original submission, we conducted ablation studies on the impact of multimodal noise, described in Section 4.2 (paragraph titled "From unimodal to multimodal noise through tuning") and illustrated in Figure 5. These results demonstrate that multimodal noise (including the tuning technique) improves generation quality by preventing class-blending artifacts.
> Additionally, we have now included (Appendix D.5) an analysis of the development of the entanglement structure across the generator layers, quantified via subsystem entropies. This analysis shows how the generator explicitly leverages multimodality, as reflected in the clearly multimodal distribution of entropies, providing further support for its beneficial role.
> We hope these clarifications make the contribution of multimodal noise fully transparent, and we appreciate the referee’s helpful suggestion.
>
> > What quantitative metrics were used to assess image quality and diversity? Including standard benchmarks like FID or IS would strengthen your claims and allow comparison with classical GANs.
>
> We highly appreciate the referee's question. We believe that following the referee's suggestion of including FID metrics to provide a quantitative comparison substantially strengthens our revised work. While we note certain limitations regarding the comparability of FID scores - see Appendix D.1 (e.g., dataset- or subset-dependence, susceptibility to image compression artifacts) - we explicitly provide comparisons with both prior quantum and classical GAN works in the Discussion (Section 5). This allows us to both demonstrate significant improvements over previous (comparable) QGAN baselines and contextualize our work within the broader classical machine learning landscape.
> We would like to emphasize, however, that our goal is not to outperform classical methods or demonstrate quantum advantage. We fully acknowledge that modern classical generative models, which often differ from the standard GAN setting, e.g., diffusion models, are far more mature than quantum approaches, including ours. Instead, our contribution lies in advancing quantum generative approaches and progressively closing the gap to classical performance in the long term.

---

> ### Author Response · Authors · 2025-11-20
> **Official Response to Reviewer u5rB (part 4)**
>
> ### Questions (cont'd):
>
> > How feasible is your approach on current noisy intermediate-scale quantum (NISQ) hardware? Please provide resource estimates (e.g., qubit count, circuit depth) and discuss any hardware-specific constraints.
>
> We thank the referee for this critical question, which directly addresses the very timely topic of NISQ applicability. Explicit resource estimates for our generator circuits, including compiled circuit depths and CNOT counts, are now provided in Appendix B.5 (Table 1). These estimates follow standard compilation procedures over a gate set of single-qubit Pauli rotations and two-qubit CNOT gates, with circuit depth defined as the maximum number of non-parallelizable gates and CNOT counts reported as fractions of total gates. We note that, as expected, the models exceed the capabilities of current NISQ hardware due to their circuit depth beyond typical coherence times. Because of this already, no real-device benchmarks or hardware-noisy simulations were conducted. Our target hardware was not intended to be NISQ devices. Instead, the work is aimed at near-term fault-tolerant systems (see lines 478f in the revised manuscript). The relatively low qubit count (11–13) makes these circuits particularly interesting for such systems, which are expected to become available in the near future - especially in contrast to many fault-tolerant quantum algorithms that require qubit numbers orders of magnitude higher. The resource estimates provided thus offer a concrete guide for assessing feasibility on near-term fault-tolerant devices.

---

> ### Comment · Reviewer_u5rB · 2025-11-26
>
> Thanks for the reply. The authors partially solve my questions, and I would like to keep my score.

---

> > ### Author Response · Authors · 2025-11-27
> > **Authors’ Follow-Up to Reviewer u5rB**
> >
> > **Authors’ Follow-Up to Referee u5rB**
> >
> > We thank the referee again for the careful initial review and the update acknowledging our rebuttal. We truly appreciate the time invested and are grateful for the constructive feedback that helped us improve the manuscript substantially.
> >
> > The referee's update says that our responses “partially” solved their questions while keeping the original score of 4. We respectfully believe the major revisions uploaded on Nov 20 thoroughly resolve all of the points raised. To make this easy to verify, here is a concise mapping:
> >
> > - **Structural clarity & schematic**: Section 3 completely rewritten from the ground up; Fig. 1 extended into a full modular overview of the QGAN workflow (initial noise ↔ generator ↔ discriminator & data).
> > - **Novelty vs. Chakrabarti et al. (2019) & Tsang et al. (2023)**: The initial Wasserstein QGAN (Chakrabarti et al., 2019) trageted quantum data generation, while we concern classical data and, hence, build upon Tsang et al. (2023), we added an explicit discussion in Section 3 and lines 152–154; the novelty of our work stems from our application-specific generator, which enables high-quality full-resolution (not patch-based or classical dim. reduction) image generation on 10-class datasets: a significant improvement over previous state-of-the-art baseline (Tsang et al., 2023), as (quantitatively) discussed in lines 486-491 and (qualitatively) detailed in Appendix D7.
> > - **Reproducibility & experimental details**: Full codebase including explicit configurations of all experiments was provided as supplemental material in initial submission; Appendix B contained information on tools (PennyLane + JAX state-vector simulation) and hyperparameters, but extended further (e.g., parameter counts; Appendix B, Table 1). Concerns about reproducibility and transparancey should be minimal.
> > - **Circuit diagrams**: Application-specific circuit diagram present in original manuscript (Fig. 1); added explicit cross-reference to gate definitions in Appendix A (line 158) and agnostic baseline circuit diagram in Appendix B.2.1 (Fig. 8).
> > - **Unimodal vs multimodal noise**: The requested ablation study was already included in the original main text (see Section 4.2 and Fig. 5). It has been further enhanced with a new analysis of entanglement entropy (Appendix D.5, Fig. 14), which highlights the differences in the entanglement structure generated based on the input noise mode. This underscores the significance of this novelty.
> > - **Quantitative metrics**: As explicitly requested by the referee, we added FID for numerous experiments throughout Section 4 (and Appendix D.3). Importantly, this allows for quantitative comparison with classical WGAN-GP and quantum (Tsang et al., 2023) baselines (see Discussion lines 482-506, and Appendix D.7).
> > - **Resource estimate & hardware feasibility**: New resource estimates (compiled CNOT count and depth estimates; Appendix B.5, Table 1) + explicit discussion (lines 478ff) that the method targets near-term fault-tolerant devices (11–13 logical qubits, typical depth beyond NISQ).
> >
> > These changes directly respond to every weakness and question the referee listed, and we believe they transform the earlier concerns about clarity, reproducibility, novelty, and relevance.
> >
> > We would be very grateful if the referee could take a quick look at the revised version again and let us know whether, in light of these concrete additions, the remaining concerns are resolved and a score adjustment feels warranted. Otherwise, we would invite the referee to share open concerns or questions that justify the current score, and we are willing to provide a new revision of our manuscript.
> >
> > We thank the referee once again for their expertise and for helping us strengthen the paper. We are happy to add any further details and clarifications.

---

> > > ### Comment · Reviewer_u5rB · 2025-11-27
> > >
> > > Thank you for your reply. If this were a journal article, I would be very happy to accept it after you have made comprehensive revisions based on my suggestions. However, since this is a conference paper, the initial presentation did not allow me to fully understand your model, and the quality of the first draft has largely determined the evaluation result. The paper is somewhat lacking in its articulation of the innovative points, failing to achieve conciseness and clarity. I suggest that you further strengthen the direct presentation of the core innovations when submitting to top-tier conferences in the future. I understand that fully expressing the innovative points within a limited space is indeed challenging, but it is also crucial for the success of conference papers.

---

### Official Review · Reviewer_o4r5 · 2025-11-06

**Soundness:** 2
**Presentation:** 3
**Contribution:** 2
**Rating:** 2
**Confidence:** 4

**Summary:**

The paper proposes a quantum Wasserstein GAN (QWGAN) capable of generating full-resolution MNIST, Fashion-MNIST, and SVHN images using a task-specific variational quantum circuit aligned with FRQI/MCRQI encodings. The key contributions include (1) an architecture with inductive bias derived from hierarchical pixel indexing and FRQI structure, (2) a multimodal, learnable noise injection mechanism intended to improve sample diversity, and (3) experiments showing high-quality image generation in numerical simulation. The paper includes ablations on ansatz design, noise modeling, overmoding, and finite-shot effects.

**Strengths:**

- The paper identifies and addresses a primary challenge in QML: the inability of most quantum models to handle high-dimensional classical data directly.
- The ablation between task-agnostic vs. FRQI-aligned circuits is convincing and highlights the importance of structured ansatz in QML. The ablation study in Figure 4 successfully demonstrates that a circuit ansatz specifically designed for the FRQI encoding produces higher-quality images than a generic ansatz, though this result is somewhat expected.

**Weaknesses:**

- All results are purely classical simulations, making claims of scalability speculative. The proposed models use 11-13 qubits and 64-layer deep circuits with >10k parameters, which are far beyond the capabilities of NISQ hardware. No evidence is provided that these circuits can be trained or executed on real devices or even on moderate-scale noisy simulators. No evidence is provided that these circuits are not suffering from the barren plateau phenomenon.

- The entire premise of the paper's "scaling" advantage is built on the FRQI encoding, which compresses an $N$-pixel image into $O(\log N)$ qubits. While the authors celebrate this exponential compression in storage, they fail to address the well-known and fatal flaw of this method: recovering the $N$-pixel image requires $O(N)$ measurements. This exponential cost in runtime (measurement) completely negates any benefit from the qubit compression. A method that scales exponentially with the problem size cannot, by definition, be considered "scalable." The authors relegate this fundamental issue to a brief discussion of "future work" (e.g., compressed sensing, shadow tomography), which is unacceptable. This is not a minor detail to be "left for future work"; it is a core flaw that invalidates the paper's central claim of having achieved a scalable solution.

- The paper compares only against previous quantum baselines. There is no evaluation or any metric that would place quantum performance relative to classical GANs of similar size. The absence of such baselines makes it difficult to judge the relevance of the results for the broader machine learning community. No evidence is provided that the quantum model would outperform the classical models.

**Questions:**

- Given the hybrid setup, isn't the classical CNN discriminator doing almost all of the intelligent work? How can you be certain that the quantum generator is anything more than a complex, over-parameterized noise source that the powerful classical discriminator simply learns to interpret? How does performance compare quantitatively to classical GANs with a similar number of parameters?

- The "future work" proposals for the measurement problem are purely speculative. Can you provide any concrete evidence or preliminary data that methods like compressed sensing would actually work? For many natural images, the sparsity constant $k$ is still proportional to $N$, meaning a sub-sampling approach would still be $O(N)$ and offer no real savings.

- The literature review is somehow inadequate, e.g., the very first paper that proposed quantum WGANs (https://arxiv.org/abs/1911.00111) is not even mentioned here.

---

> ### Author Response · Authors · 2025-11-20
> **Official Response to Reviewer o4r5 (part 1)**
>
> We express our gratitude to the referee for their time and helpful feedback on our work. We are glad the referee recognized several convincing strengths in our work. In light of the identified weaknesses and questions raised by all referees, we provide a revised version of our work. Before directly addressing the referee's specific comments, we present a list of major modifications:
>
> - Ground-up revision of Method (Section 3) plus expanded modular schematic (Fig. 1).
> - Added quantitative evaluation via Fréchet Inception Distance (FID) metric for relevant experiments
> - Added benchmarks/comparisons: both quantum (patch-generation QGAN (Tsang 2023)) and classical (WGAN-GP), see Discussion (Section 5)
> - Added inductive bias rationale theoretically (Appendix B.2.2) and numerically (Appendix D.5)
> - Added new analysis on gradient scaling with image size, comparing behavior for image-specific vs agnostic generator design (Appendix D.6).
> - Added model requirements (incl. parameter counts) and quantum resource estimates (compiled circuit depths and CNOT counts) in Appendix B.5, Table 1 for the different generator configurations used.
>
> We continue to address each point regarding weaknesses and questions the referee explicitly raised. We first recite the original statement, then provide the reply.
>
> ## Weaknesses:
>
> > - All results are purely classical simulations, making claims of scalability speculative. The proposed models use 11-13 qubits and 64-layer deep circuits with >10k parameters, which are far beyond the capabilities of NISQ hardware. No evidence is provided that these circuits can be trained or executed on real devices or even on moderate-scale noisy simulators. No evidence is provided that these circuits are not suffering from the barren plateau phenomenon.
>
> We thank the referee for raising their concerns regarding the applicability to current hardware. It is true that the models exceed the capabilities of current NISQ hardware due to the circuit depth (see newly added Table 1 in Appendix B.5). Because of this expectation, no real device benchmarks or (hardware) noisy simulations were conducted. The target hardware of this work was not claimed to be NISQ devices. Instead, fault-tolerant hardware should be seen as the target (we add a statement to the Discussion, Section 5, lines 478f). The relatively low (11-13) qubit count makes our work particularly interesting for near-term fault-tolerant systems, which are expected to become available in the near future, in stark contrast to many fault-tolerant quantum algorithms with qubit requirements orders of magnitude higher. To support the general applicability of our method to real (fault-tolerant) hardware, we already included (simulated) experiments with finite shots in our original submission (see Sec. 4.3 and Fig. 7). With shot noise being inevitably present even in fault-tolerant hardware and the source of barren plateaus (by dominating the concentrated quantity being estimated), the demonstrated shot noise robustness in these experiments empirically validates both an expected success for fault-tolerant hardware applicability in scaling to the full resolution of the studied image datasets. We would like to draw particular attention to Fig. 7a, where the finite-shot training shows a clear tendency toward a uniform distribution over the (marginal) pixel probabilities, i.e., observing information for any pixel upon a measurement is equally likely. Therefore, a number of measurements proportional to the number of pixels $N$ suffices to decode all pixels to a sufficient precision, i.e., estimate each pixel with a variance up to $N/T$ for a total measurement shot budget of $T$. Clearly, a non-uniform pixel distribution would lead to suboptimal estimates for some pixels, as they would be observed less often, up to extreme concentration effects where certain pixels would never be measured under realistic shot budgets.
>
> With the previous statement on concentration effects being already related to the barren plateau phenomenon, we more directly address the referee's concern regarding the emergence of such with further experiments and systematic empirical analyses in Appendix D.6. Here, we study the gradient magnitude per parameter as the image size (and, hence, the number of qubits) increases. We do so for both our proposed task-specific generator design and the agnostic baseline. This is relevant as potential trainability issues (e.g., related to barren plateaus) may arise as problem (i.e., image) size increases. We identify significantly more desirable scaling behavior in our specific generator design over the agnostic one. While this gives some preliminary empirical evidence, more rigorous barren plateau studies would require a theoretical treatment. It should be noted that the notion of barren plateaus is nuanced here, because under amplitude-type encoding (such as FRQI), the number of qubits grows only logarithmically with the problem size.

---

> ### Author Response · Authors · 2025-11-20
> **Official Response to Reviewer o4r5 (part 2)**
>
> ### Weaknesses (cont'd)
>
> > - The entire premise of the paper's "scaling" advantage is built on the FRQI encoding, which compresses an $N$-pixel image into $O(\log N)$ qubits. While the authors celebrate this exponential compression in storage, they fail to address the well-known and fatal flaw of this method: recovering the $N$-pixel image requires $O(N)$ measurements. This exponential cost in runtime (measurement) completely negates any benefit from the qubit compression. A method that scales exponentially with the problem size cannot, by definition, be considered "scalable." The authors relegate this fundamental issue to a brief discussion of "future work" (e.g., compressed sensing, shadow tomography), which is unacceptable. This is not a minor detail to be "left for future work"; it is a core flaw that invalidates the paper's central claim of having achieved a scalable solution.
>
> The concern arises from a misunderstanding of what exponential scaling occurs in FRQI encodings. Recovering an $N$-pixel image indeed requires $O(N)$ measurements, but this cost is not exponential in the problem size, which is here the number of pixels $N$, not the number of qubits $n$. Instead, it is exponential in the number of qubits, precisely because the qubit count $n$ is $O(log(N))$ to store $N$ pixels in FRQI. Thus, the measurement cost cancels the exponential compression provided by the encoding, but it does not introduce a detrimental exponential overhead relative to the classical image/problem size $N$. This characteristic is inherent to all amplitude-type encodings and is not unique to our method. Overall, we neither claim nor imply any quantum (exponential) speedup from the encoding or the measurement process. The three proposed decoding strategies are likewise not intended to provide any exponential speedup. Their role is purely to reduce measurement overhead and improve decoding robustness in practical settings. They are not essential to our scalability arguments and can therefore be reasonably deferred to future work. We thank the referee for pointing out the lack of clarity and have revised the corresponding part in the Discussion (Section 5).
>
> Furthermore, in our paper, “scaling” does not refer to a scaling advantage over classical methods. Rather, it denotes the ability to train a single, end-to-end quantum generator on full-resolution images and full dataset classes—something not previously demonstrated without classical dimensionality reduction or patch-based generation. From a complexity-theoretic viewpoint, this use of "scaling" remains consistent with standard notions, as our approach remains at most polynomial in the problem size $N$.
>
> Finally, we emphasize that quantum generative models have use cases beyond classical image reconstruction - most notably as synthetic quantum data loaders for downstream quantum algorithms. In such scenarios, the generated quantum states are consumed directly in quantum form, and full classical readout is not required. We have added a paragraph in the Discussion (Section 5) pointing to that fact more explicitly.
>
>
> > - The paper compares only against previous quantum baselines. There is no evaluation or any metric that would place quantum performance relative to classical GANs of similar size. The absence of such baselines makes it difficult to judge the relevance of the results for the broader machine learning community. No evidence is provided that the quantum model would outperform the classical models.
>
> We thank the referee for raising this point and agree that it is crucial to contextualize our QGAN framework within the broader classical machine learning community.
> Therefore, in the Discussion (Section 5) of the revised manuscript, we now relate our QGAN results quantitatively (via the Fréchet Inception Distance (FID) metric) to the classical analogous gradient-penalized Wasserstein-GAN. Here, we rely on a large-scale benchmark by Lucic et al. (2018) and their reported scores for MNIST and Fashion-MNIST.
> We conclude our response with two closing remarks on comparisons with classical generative models.
> First, we emphasize that the goal of our work is not to outperform classical approaches or to demonstrate quantum advantage, which is well beyond the scope of our work. Second, modern classical generative models (e.g., diffusion models) are more capable and more mature than current quantum approaches, including our work. To improve the clarity of the goals of our work, both statements are explicitly added to the Discussion (Section 5).
> Hence, the relevance of our work lies in advancing quantum generative modelling to successively close the gap with classical ML, which is essential in the long term.

---

> ### Author Response · Authors · 2025-11-20
> **Official Response to Reviewer o4r5 (part 3)**
>
> ## Questions:
>
> > - Given the hybrid setup, isn't the classical CNN discriminator doing almost all of the intelligent work? How can you be certain that the quantum generator is anything more than a complex, over-parameterized noise source that the powerful classical discriminator simply learns to interpret? How does performance compare quantitatively to classical GANs with a similar number of parameters?
>
> We thank the referee for raising these questions and the underlying concern about how generative responsibility is divided between classical and quantum components. The issue arises from a misinterpretation of the QGAN workflow: the classical CNN discriminator aids training only and plays no role during inference. It is discarded after training, and the quantum generator alone produces new samples. The generator is therefore not a mere noise source but learns a genuine mapping from the input noise distribution to the image-data distribution. Moreover, because our generator is fully end-to-end and involves no classical post-processing (unlike trick #1), its generative performance cannot be attributed to classical components. To clarify this, we improved the presentation of our QGAN framework in a ground-up revision of Section 3 and, in particular, refined Fig. 1 to make the components and their interplay more explicit.
>
> > - The "future work" proposals for the measurement problem are purely speculative. Can you provide any concrete evidence or preliminary data that methods like compressed sensing would actually work? For many natural images, the sparsity constant $k$ is still proportional to $N$, meaning a sub-sampling approach would still be $O(N)$ and offer no real savings.
>
> We thank the reviewer for raising concerns about the speculative nature of our proposed future work involving compressed sensing. We address the points below:
> 1. We would like to reiterate that our quantum machine learning model does not incur exponential measurement overhead. Instead, our proposed methods maintain a polynomial complexity in the problem size N.
> 2. We acknowledge that we do not currently have preliminary data for the proposed compressed sensing approach. This is precisely why it is outlined as future work, i.e., as a promising direction that we plan to explore further. We included this direction to highlight the potential of leveraging structured sparsity in the measurement process.
> 3. We have considered two complementary strategies for applying compressed sensing to the measurement problem, both grounded in established theoretical frameworks:
> __(a)__ Classical Compressed Sensing as a Denoising Post-Processing Step: In this method, the output of the quantum model is measured in the computational basis O(klog(N/k)) times, where k is the sparsity level of the signal. We then apply a classical LASSO-type regression or similar convex optimization method to reconstruct a denoised version of the signal. This approach is related to the work of Candes and Wakin [Candes & Wakin, IEEE Signal Processing Magazine, 2008], and has been successfully applied in compressed sensing scenarios where the signal exhibits sparsity or compressibility. We expect this approach to yield good performance when the image has a sparse representation in some known basis.
> __(b)__ Quantum Tomography via Compressed Sensing: Alternatively, we consider applying quantum compressed sensing techniques to perform state tomography. Specifically, we would measure the quantum state over an informationally complete basis O(kN(logN)^2) times, as demonstrated by [Gross et al., Phys. Rev. Lett., 2010]. Once the full quantum state is reconstructed, the classical image can be retrieved by squaring the probability amplitudes. While this method has a higher sample complexity than the classical approach, it leverages the sparsity of the underlying quantum state and offers a more general recovery path that may prove effective in practice.
>
> We hope this clarifies our rationale and grounds the proposed ideas in existing, peer-reviewed frameworks. We appreciate the reviewer’s attention to the speculative nature of future directions and have made sure to emphasize the distinction between our current contributions and our outlook for subsequent work.

---

> ### Author Response · Authors · 2025-11-20
> **Official Response to Reviewer o4r5 (part 4)**
>
> ### Questions (cont'd):
>
> >  The literature review is somehow inadequate, e.g., the very first paper that proposed quantum WGANs (https://arxiv.org/abs/1911.00111) is not even mentioned here.
>
> We appreciate the referee for their careful reference to the existing literature. Indeed, the highlighted work first introduces the Wasserstein distance in the quantum GAN setting, analogous to classical Wasserstein-GANs. In order to present a focused literature review relevant to our application, however, we restricted it to quantum generative models for image generation. Hence, this work was omitted not due to an inadequate completeness of the literature review, but because it deviated in application, i.e., quantum data rather than classical image generation.
> Our work is most closely related to Tsang et al. 2023 in terms of both the image application and Wasserstein QGAN extension, which was introduced before for classical data by Herr et al. 2021. Nevertheless, we value the reviewer's hint and believe that including these details and references in our paper improves the clarity on how our work is placed in the existing body of literature (see lines 152-154).

---

> ### Comment · Reviewer_o4r5 · 2025-11-27
>
> I thank the authors for the detailed responses that partially address my concern, particularly in the definition of "scaling". However the main concern on the missing quantum improvement and impact to the broader machine learning community still remains. I hereby raise my score to 4.

---

### Author Response · Authors · 2025-12-03
**Relevance and contribution to the field**

Quantum machine learning is still in its early stages, where responsible progress comes from rigorous, incremental advances rather than unrealistic claims of quantum advantage over classical methods. Systematic empirical studies on realistic data, demonstrating improvements over prior quantum methods, are essential to maturing the field and bridging the gap to classical ML. Such contributions are scientifically valuable and complementary to more abstract theoretical work.

Our work contributes to this goal by achieving substantial improvement over prior quantum approaches, showing limited quality despite operating on low-resolution image patches and only up to three image classes, through principled design choices that introduce inductive bias tailored for image generation.

---

### Meta-Review · Area_Chair_C7wf · 2025-12-27

**Summary:**

This paper applied the quantum Wasserstein GAN (QWGAN) for generating full-resolution MNIST, Fashion-MNIST, and SVHN images using a task-specific variational quantum circuit aligned with FRQI/MCRQI encodings. Experiments covered generating samples of high quality and diversity as well as ablations on ansatz design, noise modeling, overmoding, and finite-shot effects.

In the reviews, the reviewers raised concerns about:
- Scalability issue: There is no real device benchmarks or (hardware) noisy simulations, and the proposed models applied with >10k parameters, which are far beyond the capabilities of NISQ hardware.
- Comparisons with classical benchmarks were lacked, and it's not clear whether quantum achieves advantage for image classification.
- Lack of theoretical analysis.

During the rebuttal, the authors made various clarifications about these points. However:
- The title with "high-resolution" is misleading, as the pictures used in Section 4 has 28*28 or 32*32 pixels, extremely far from the normal sense of high-resolution;
- On the theory side, there is no evidence that the proposed model has advantage over classical counterparts, and it's hence unclear whether this work is helpful in the long term. During the rebuttal the authors added in Appendix D.5 that the development of entropy in different subsystems, but this is not a direct evidence about whether the model is good or not.
- On the experimental side, the QWGAN model per se had been proposed in earlier works, and the improvements claimed by this paper were not implemented on real quantum devices. The authors also admitted that the target hardware of this work was not claimed to be NISQ devices, making the importance of this work unclear in the short term.

In all, given these points and the dominating negative scores, the decision is rejection at ICLR 2026.

**Reviewer Concerns:**

Some details were clarified by the authors' rebuttal, but there are still outstanding issues as mentioned in the meta-review.

**Reviewer Scores:**

I don't think this question applies to this paper by much - there were adequate discussions between the authors and the reviewers.

---

### Decision · Program_Chairs · 2026-01-26

Reject